# Towards Physics-informed Spatial Intelligence with Human Priors: An Autonomous Driving Pilot Study

**Guanlin Wu**[*]   **Boyan Su**[*]   **Yang Zhao**   **Pu Wang**   **Yichen Lin**   **Hao Frank Yang**[†]

Johns Hopkins University

{gwu32, haofrankyang}@jhu.edu

Project page: https://guanlinwu123.github.io/sigbench

## Abstract

How to integrate and verify spatial intelligence in foundation models remains an open challenge. Current practice often proxies Visual-Spatial Intelligence (VSI) with purely textual prompts and VQA-style scoring, which obscures geometry, invites linguistic shortcuts, and weakens attribution to genuinely spatial skills. We introduce **Spatial Intelligence Grid (SIG)**: a structured, grid-based schema that explicitly encodes object layouts, inter-object relations, and physically grounded priors. As a complementary channel to text, SIG provides a faithful, compositional representation of scene structure for foundation-model reasoning. Building on SIG, we derive SIG-informed evaluation metrics that quantify a model's intrinsic VSI, which separates spatial capability from language priors. In few-shot in-context learning with state-of-the-art multimodal LLMs (*e.g.* GPT- and Gemini-family models), SIG yields consistently larger, more stable, and more comprehensive gains across all VSI metrics compared to VQA-only representations, indicating its promise as a data-labeling and training schema for learning VSI. We also release `SIGBench`, a benchmark of 1.4K driving frames annotated with ground-truth SIG labels and human gaze traces, supporting both grid-based machine VSI tasks and attention-driven, human-like VSI tasks in autonomous-driving scenarios.

## 1 Introduction

Currently, Visual Question Answering (VQA) is recognized as a natural test of visual-spatial intelligence (VSI), as it requires interpreting images, understanding object relationships, and inferring context to produce correct answers using text [1–19]. In computer vision, most researchers agree that VSI involves accurately perceiving, manipulating, and reasoning about visual and spatial information, such as size, location, and correlations given an input image, using textual description [20]. In a state-of-the-art study, three VQA questions appear on the first page that illustrate their VSI understanding: *"What is the distance between the keyboard and the TV? How many cabinets are in the room? And how height the stool is ?" [13]* And correct answers indicate effective VSI learning by the model.

**However, are VQA tasks ideal for evaluating spatial intelligence?** For human beings, it is defined by Dr. Gardner and Dr. Lohman as *"the ability to generate, retain, retrieve, and transform well-structured visual images"* [21]. In computer vision, compared to other well-explored vision problems, VSI's core challenge lies in extracting 3D geometric insights from 2D images or videos and then presenting them textually [13, 22]. VQA inherently interweaves linguistic proficiency with spatial reasoning. However, even without visual input, humans can form a mental map of their surroundings by relying on auditory cues, like the way sound reverberates in a space [23]. Some researchers argue that VQA's reliance on textual replies may not fully capture the underlying spatial intelligence [24–27].

---

[*]Equal contribution, [†]Corresponding author.

They advocate integrating additional physical priors or alternative more spatial-like representations to reflect spatial reasoning, like how human beings think.

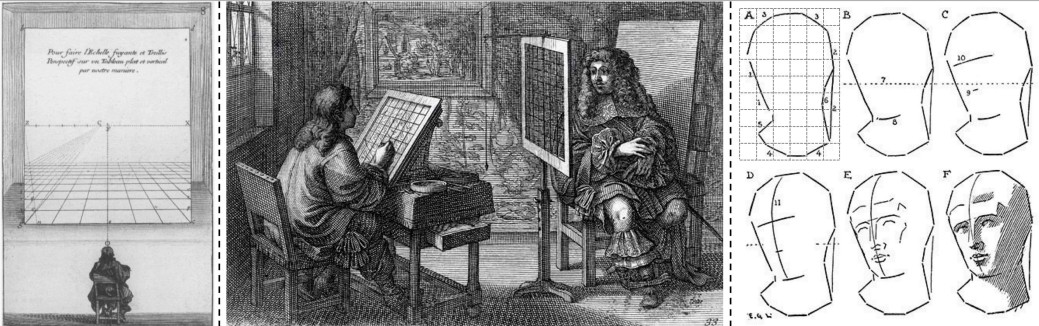

Figure 1: **Examples of Human VSI in Painting.** Abraham Bosse, a French artist and theorist illustrates a systematic, grid-based method for achieving visual-spatial correlations in rendering three-dimensional space on a two-dimensional canvas (left) [28], and he employed this grid-based visual scheme in portrait painting (middle) [29, 30]. Procedures of drawing a cast with graphical priors from scratch (right). [31]

Historical art practices offer an inspiring perspective: a grid-canvas is very helpful for humans to perceive the spatial priors. During the Renaissance, artists like Albrecht Dürer, Leonardo da Vinci and Vincent van Gogh famously used grid-based "drawing machines," as often depicted in historical artworks [32–36]. By imposing visual observations onto structured grids (see Fig. 1), artists could systematically decompose a 3D scene into spatial regions, then record or interpret its geometry and graphical correlations. As illustrated in Fig. 1 middle, portrait painters often proceeded from these organized graphical representations to finely detailed textual descriptions, mirroring how human VSI naturally moves from a structured visual framework to richer linguistic content. Consequently, once trained to perceive spatial relationships accurately for painters, the specific subject matter becomes less significant: any subject, regardless of its nature, is ultimately perceived as a unique combination of graphs (nodes, shapes, edges, and color notes) on these grids. Collectively, these visual graphs define the appearance of objects such as a cast, a flower, or a human head [37].

**Can graphical priors on a grid-based canvas be used as machine representations of VSI?** Like a painter's trained eye, machines with grid or graph-based abstractions gain a structured view that captures spatial relationships and hierarchies. This simplifies scene decomposition and supports richer interpretation and language grounding. From our perspective, an ideal VSI representation is a combination of a structured conceptual model of a scene that encodes geometric and topological relationships, typically through grid- or graph-formatting priors. In this framework, visual elements (*e.g.* shapes, edges, etc.) are mapped onto discrete spatial partitions and connected to capture both local and global information. By systematically organizing visual data into these relational structures, such an ideal VSI representation supports robust reasoning, efficient spatial manipulation, and a natural pathway from pure visual analysis to higher-level semantic or linguistic descriptions.

Building upon these insights, we propose a novel VSI representation format, called *spatial intelligence grid* (SIG), and then conduct foundational testing on existing Multimodal Large Language Models (MLLMs) in one of the most critical VSI application tasks: autonomous driving (AD) [38–42]. AD demands real-time perception, precise modeling of multiple dynamic objects, and a high level of situational awareness in diverse environments. The ability to efficiently attend to elements locations, correlations and movements in the driving scene, such as other vehicles, pedestrians and road signs is essential for safe and effective operation. Given the complexity, flexibility, high stakes, and the potential impacts, AD serves as an ideal stress test for evaluating and refining SIG-based VSI frameworks. To facilitate this, we introduce a novel benchmark, `SIGBench`, and systematically investigate three fundamental tasks: *1) SIG-based VSI Evaluation of MLLMs and Metrics:* Evaluate whether existing MLLMs can answer spatial queries based on SIG representations, and propose novel evaluation metrics for SIG based on graph similarity theory; *2) SIG-Empowered In-Context Learning (ICL) for VQA:* Investigate how SIG-informed VSI features enhance spatial intelligence in VQA tasks through few-shot ICL, and *3) Human v.s. Machine VSI Attention based on SIG:* Examine how human VSI attention differs from machine-based approaches within SIG, and explore methods to achieve more human-like SIG in order to improve human–machine interactions. In this research, our contribution can be summarized as follows:

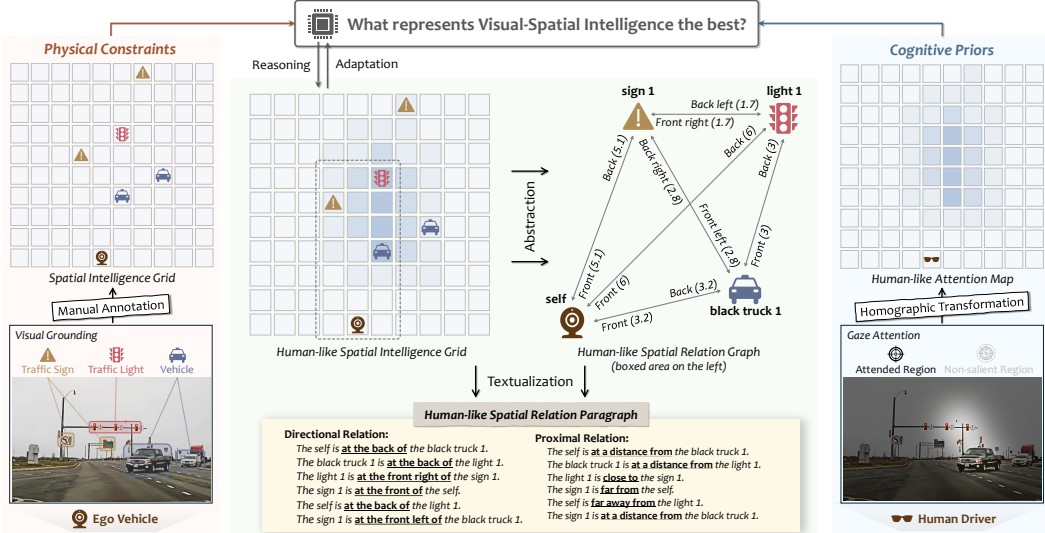

Figure 2: **Overview of Human-like SIG in AD Scenario.** In the left, we use SIG to represent the spatial relation of traffic sign, traffic lights, vehicles and self (ego-vehicle) in the image. We apply homographic transformation to convert human gaze attention from image to SIG size in the right. Combining them, we get human-like SIG, which can then be extracted to human-like SRG and SRP in middle part. The order denotes the rank of an object of the same category in the image from left to right (*e.g.* black truck 1 is the left-most object among vehicles).

- We introduce SIG, a novel grid-based representation for VSI in AD scenario that integrates physical priors and human-like attention cues, together with a suite of structured evaluation metrics for precise spatial reasoning assessment.

- We demonstrate SIG's efficacy as a data schema for enhancing VSI through few-shot ICL on MLLMs. Compared to ICL using VQA-style representation, SIG-based ICL yields larger, more stable and more comprehensive improvements of model's VSI.

- We release `SIGBench`, a benchmark containing 1.4K frames annotated with ground-truth SIG and human gaze attention map, enabling evaluation of both grid-based machine VSI and human-like attention-driven VSI with grid under our proposed evaluation metrics.

## 2 Related Work

**General VSI in MLLMs.** MLLMs have demonstrated promising performance across a variety of visual tasks such as object detection, segmentation, and image captioning by virtue of their unified vision–language representations and strong reasoning capabilities [43–49]. Building on these strengths, recent work has extended MLLMs to tackle visual–spatial reasoning queries such as *which object is at the left-most position in this image*, through architectural adaptations and fine-tuning strategies designed to emphasize spatial relations [19, 50–56]. To provide accurate and in-depth evaluation of a model's general VSI, several comprehensive visual-task benchmarks incorporate dedicated visual-spatial reasoning sections [1–8], while others focus exclusively on assessing visual-spatial reasoning ability over images [9–12, 16–18] or videos [13–15] in VQA-style. Although these benchmarks yield clear right–wrong scores, they often conflate failures of visual grounding (*e.g.* object detection errors) with genuine visual-spatial reasoning mistakes, particularly in scenes containing many similar entities (*e.g.* multiple vehicles with different brands, colors and shapes).

**VSI with Scenario-based Physical Constraints.** In real-world applications such as robotics [62–64], AR/VR [65–67] and AD [68–70], the implementation of VSI must respect domain-specific physical, environmental and regulatory constraints. For AD in particular, models must not only interpret scene geometry but also adhere to traffic rules and kinematic feasibility across perception [71–76], planning [68, 77–80], and control modules [81–85]. A suite of VQA-style benchmarks has emerged to evaluate visual-spatial reasoning under these constraints [58–61, 86, 87], leveraging large-scale open-source AD datasets [88–92], shown in Tab. 1. However, by framing questions solely as text-image queries, these benchmarks may overlook the rich spatial structure exposed by bird's-eye-view (BEV) representation that are widely used in AD perception and planning pipelines.

Table 1: **Comparison between existing VSI and AD benchmarks.** Comp. is comprehensive. MC and AM means multiple-choice and attention map, respectively. Size means the total number of words in each dataset. Ann. and Rep. means annotate and repurpose (the dataset is compiled from prior datasets), respectively. S.R.R. means spatial relation representation. Text and Graph means whether the calculation of evaluation metrics is based on text (*e.g.* MC, sentence) or graph comparison. H-L means human-like and Auto means there are predefined metrics and can evaluate the answer automatically.

| Type | Benchmark | Benchmark Overview | | | | | Evaluation Metrics | | | |
|------|-----------|------|--------|--------|--------|--------|------|-------|-----|----------|
| | | Size | Source | #Tasks | Answer | S.R.R. | Text | Graph | H-L | A/G Eval. |
| Comp. | SEED [1] | 1.682M | Ann. | 12 | MC | Text | ✓ | ✗ | ✗ | Auto |
| | MMBench [3] | 0.096M | Rep. | 20 | MC | Text | ✓ | ✗ | ✗ | GPT |
| | MM-Vet [5] | 0.003M | Rep. | - | MC | Text | ✓ | ✗ | ✗ | GPT |
| | MMMU [6] | 0.324M | Ann. | 30 | MC/Open | Text | ✓ | ✗ | ✗ | Auto |
| VSI Only | VSR [9] | 0.094M | Rep. | 1 | T/F | Text | ✓ | ✗ | ✗ | Auto |
| | GQA [57] | 24.42M | Rep. | 1 | Open | Text | ✓ | ✗ | ✗ | Auto |
| | SRBench [10] | 0.068M | Rep. | 4 | MC | Text | ✓ | ✗ | ✗ | Auto |
| Comp. AD | NuScenes-QA [58] | 6.592M | Rep. | 5 | Open | Text | ✓ | ✗ | ✗ | Auto |
| | NuScenes-MQA [59] | 43.43M | Rep. | 4 | Open | Text | ✓ | ✗ | ✗ | Auto |
| | NuPlanQA [60] | 16.89M | Rep. | 9 | Open | Text | ✓ | ✗ | ✗ | Auto |
| VSI AD | NuScenes-SpatialQA [61] | 78.75M | Rep. | 2 | MC/Open | Text | ✓ | ✗ | ✗ | Auto |
| | SIGBench | 71.35M | Ann. | 5 | MC/SIG/AM | SIG | ✓ | ✓ | ✓ | Auto |

**Human-Like VSI.** Human scene understanding is guided by both overt gaze patterns (*where we look*) and covert cognitive maps (*how we represent spatial relations*). Gaze or saliency prediction models, which estimate pixel-level attention maps, have provided deep insights into visual prioritization in generic images [93–97] and driving scenarios [98–101]. Complementing this, the notion of a cognitive map captures the internal spatial layout that humans use to reason about object relations [102] and has recently been integrated into MLLM frameworks for video understanding [13]. In our approach, we leverage an human-like SIG to emulate both gaze-driven saliency and structured spatial representations, thereby enabling a human-like evaluation of visual-spatial reasoning, particularly within complex AD environments.

## 3 Methods

### 3.1 Grid-based Visual-Spatial Intelligence

We introduce SIG, a grid-based representation format for VSI, as illustrating in Fig. 2. In AD scenarios, roadways will be a physical constraints and the primary entities we focus on are the ego-vehicle and other traffic nodes, scuh as vehicles, traffic signs, signal lights and traffic lanes. Based on SIG, we can extract a directed spatial relation graph (SRG) that describes the spatial relation (direction+distance in grid) of each object and spatial relation paragraph (SRP) that describes spatial relation of each object within a text manner. To quantitatively assess a model's VSI, we propose three novel evaluation metrics: multi-level spatial matching (MLSM), spatial relation graph similarity (SRGS) and semantic relational distance (SRD). MLSM compares object positions directly within the SIG representation, capturing absolute localization accuracy. SRGS measures both node-wise and edge-wise correspondence between predicted and ground-truth (GT) SRG, emphasizing relation classification and structure. SRD computes a semantic relational distance between predicted and GT prepositions in SRP, evaluating the fidelity of both directional and proximal relations. To isolate core spatial reasoning, our evaluation focuses exclusively on ego-vehicle (self), other vehicles, traffic lights, and traffic signs, omitting traffic lanes from the quantitative metrics because many scenes have unreliable lane cues such as faded or blurred markings and no markings at intersection or rural roads (we provide evaluation metrics including traffic lanes for an additional subset with clear traffic lanes). Before evaluating predicted SIG using GT SIG, we will firstly align the position of self in predicted SIG to which in GT SIG. Next, we apply the same offset for all other objects in predicted SIG.

**Multi-level Spatial Matching.** To evaluate SIG-based spatial representations independently of visual-grounding noise, we introduce a multi-threshold graph matching protocol inspired by tracking metrics such as MOTA [103] and HOTA [104]. MLSM first performs bipartite matching between predicted and GT objects (vehicles, traffic signs, traffic lights) using a cost function $c_v$ for vehicles and $c_{sl}$ for traffic signs and traffic lights, which can be calculated by

$$c_v = d * \omega_c * \omega_o * \omega_t, \quad c_{sl} = d * \omega_o \tag{1}$$

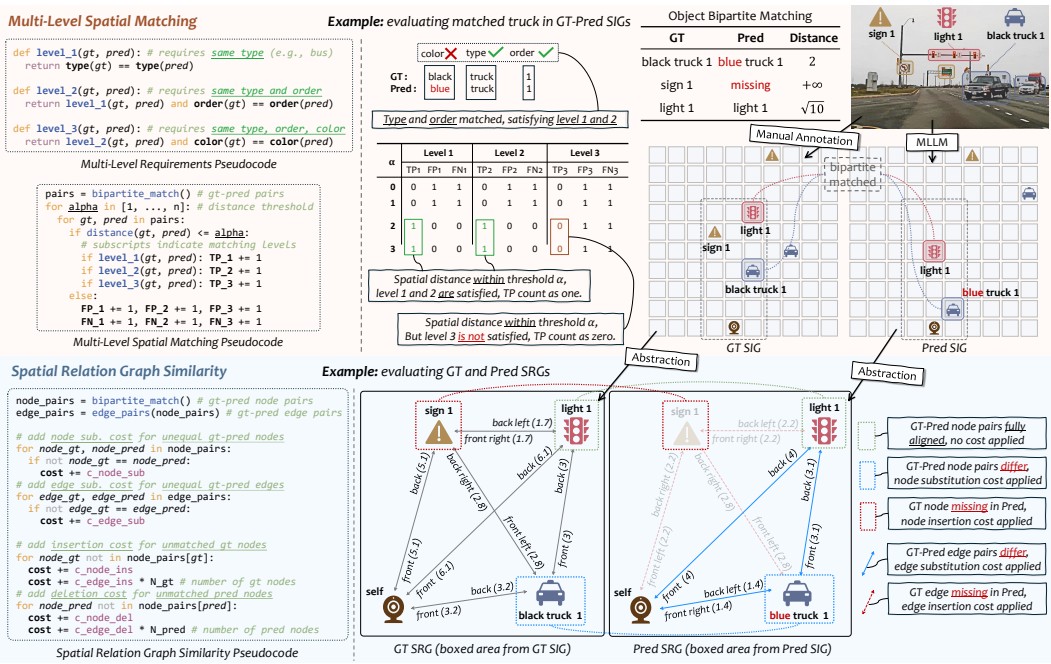

Figure 3: **Illustration Examples of MLSM and SRGS.** At the start of both MLSM and SRGS, it will match the objects between the predicted and GT SIG using bipartite matching. For MLSM, we provide example for calculating TP, FP and FN for vehicles in the boxed area in upper part. For SRGS, we highlight the node and edge needed for insertion and substitution and their total edit distance in lower part.

where $\omega_c, \omega_o, \omega_t$ means the weight for color, order and type matching, respectively. Here $d$ is the euclidean distance between objects' position on SIG, and each weight $\omega_c, \omega_o, \omega_t$ equals one when the attribute is unmatched and drops below one when it matches, thus granting larger spatial tolerances for objects that are matched in type, order, and/or color.

After matching, true positives (TP) are object pairs whose grid position lie within a distance threshold $\alpha$ and object attributes are aligned. False negatives (FN) are GT objects with no predicted match; false positives (FP) are predicted objects with no GT match. We evaluate three hierarchical matching levels for vehicles: (1) same type, (2) same type + order, (3) same type + order + color, and one level for signs and lights (same order), respectively. Over a set of $n$ thresholds $\alpha \in [1, \ldots, n]$, we compute precision $P_\alpha$, recall $R_\alpha$, F1-score $F1_\alpha$ and association accuracy $AssA_\alpha$ and normalize them to get overall $P, R, F1$ and $AssA$, as showing in Fig. 3. Detailed calculation are shown in Appendix A.1.

Let $n$ denote the total number of objects ($n$ is used similarly in the following complexity analysis). For MLSM, constructing the cost matrix for vehicles, traffic signs, and traffic lights has complexity $O(n^2)$, bipartite matching requires $O(n^3)$ [105], and multi-threshold matching takes $O(n)$. Thus, the overall complexity of MLSM is $T_{\text{MLSM}}(n) = O(n^2) + O(n^3) + O(n) \Rightarrow O(n^3)$.

**Spatial Relation Graph Similarity.** Different from MLSM, SRGS evaluates the correspondence of individual relations (edges) between a predicted SRG and its GT counterpart. We quantify this through the computation of the graph edit distance (GED) [106] which measures the number of operations needed to edit the predicted SRG to GT. Let's denote directed SRG as $\mathcal{G} = (\mathcal{V}, \mathcal{E})$. $\mathcal{V} = \{v_i\}_{i=1}^n$ denotes the set of all $n$ nodes, where $v_i$ represents an instance (*e.g.* ego-vehicle, other vehicles, traffic signs, traffic lights). $\mathcal{E} \subseteq \mathcal{V} \times \mathcal{V}$ denotes set of edges, where directed edge $e_{ij} = (v_i, v_j) \in \mathcal{E}$ encodes spatial relation (direction + distance in SIG) from $v_i$ to $v_j$.

We further calculate the SRGS from two perspectives: node edit distance and edge edit distance. The computation is based on bipartite matching between nodes in the GT and predicted SRG. Let $M$ be the set of all matched pairs $(v_i, \hat{v}_{i'})$, where $v_i \in \mathcal{V}$ is a GT node and $\hat{v}_{i'} \in \hat{\mathcal{V}}$ is its corresponding predicted node, as determined by the bipartite matching algorithm. The node edit distance considers three types of costs: *1) Substitution cost* $\delta_{\text{sub}}(v_i, \hat{v}_{i'})$: the cost of modifying a predicted node $\hat{v}_{i'}$ to match a GT node $v_i$ with different position or attributes; *2) Deletion cost* $\delta_{\text{del}}(\hat{v}_{i'})$: the cost of removing an unmatched predicted node $\hat{v}_{i'}$; *3) Insertion cost* $\delta_{\text{ins}}(v_j)$: the cost of adding an

unmatched GT node $v_j$. The total node edit distance between the GT graph $\mathcal{G}$ (with node set $\mathcal{V}$) and the predicted graph $\hat{\mathcal{G}}$ (with node set $\hat{\mathcal{V}}$) can then be denoted as

$$D_N(\mathcal{G}, \hat{\mathcal{G}}) = \sum_{(v_i, \hat{v}_{i'}) \in M} \delta_{\text{sub}}(v_i, \hat{v}_{i'}) + \sum_{v_i \in \mathcal{V}} \delta_{\text{del}}(v_i) + \sum_{\hat{v}_j \in \hat{\mathcal{V}}} \delta_{\text{ins}}(\hat{v}_j) \qquad (2)$$

Detailed computations of each cost function are provided in Appendix A.2. Similarly, we define the edge edit distance $D_E(\mathcal{G}, \hat{\mathcal{G}})$ with the edge substitution cost $\delta_{\text{sub}}^E(e_i, \hat{e}_{i'})$, edge deletion cost $\delta_{\text{del}}^E(\hat{e}_{i'})$, and edge insertion cost $\delta_{\text{ins}}^E(e_j)$. Combining $D_N$ and $D_E$, we can calculate weighted total graph edit distance $D_{\text{total}}$ and weighted graph similarity score $S$ by

$$D_{\text{total}} = \gamma D_N(\mathcal{G}, \hat{\mathcal{G}}) + \beta D_E(\mathcal{G}, \hat{\mathcal{G}}), \quad S = \max\left(0, \ 1 - \frac{D_{\text{total}}}{D_{\text{max}}}\right) \in [0, 1] \qquad (3)$$

where $\gamma, \beta$ are weights and $D_{\text{max}}$ denotes worst-case distance (all nodes and edges unmatched).

For SRGS, constructing the fully connected graph requires $O(n)$ for all nodes and $O(n^2)$ for all edges. Bipartite matching again requires $O(n^3)$, and GED, involving node-to-node and edge-to-edge comparisons, takes $O(n) + O(n^2)$. Therefore, the overall complexity of SRGS is $T_{\text{SRGS}} = 2(O(n) + O(n^2)) + O(n^3) \Rightarrow O(n^3)$.

**Semantic Relational Distance.** To quantify the fidelity of directional and proximal preposition predictions in SRP, we assign each preposition a position on a discrete scale and measure the minimal cyclic or linear separation from GT. Directional relations are arranged cyclically, illustrated as Fig. 4. Proximal relations are ordered linearly from closest to furthest: adjacent to, close to, at a distance from, far from, far away from. Here the semantic relational distance between "adjacent to" and "far from" is 3. We compute MAE and MSE based on semantic relational distance and accuracy. Detailed equations are shown in Appendix A.3. For SRD, we compute pointwise distances. With 8 directional and 5 proximal relations, the time complexity is $T_{\text{SRD}} = O(8n) + O(5n) \Rightarrow O(n)$.

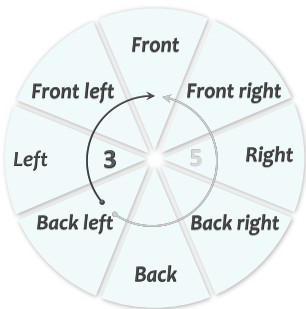

Figure 4: **Directional Relation Circle**. The semantic relational distance between any two prepositions is the smallest step count around the circle (*e.g.* between "at the back left of" and "at the front of" is 3 instead of 5).

## 3.2 SIG-based In-context Learning for VSI

While existing methods focus on improving MLLM's VSI based on depth, detection bounding box (bbox) and segmentation mask and convert them into textual representation such as QA for learning [50–52], we investigate SIG's potential as direct representation for improving VSI using ICL. Let's denote a dataset $\mathcal{D} = \{(x_i, y_i)\}_{i=1}^N$ containing $N$ image-SIG pairs. The ICL process for outputting answer $y_q$ of query $x_q$ can be formulated as

$$y_q = \mathcal{F}_M(x_q; \mathcal{P}) \qquad (4)$$

where $\mathcal{F}_M$ is the MLLM and example prompt $\mathcal{P} = \text{concat}(x_1, y_1, \ldots, x_k, y_k)$. In our settings, we randomly select $k$ image-SIG pairs where $x_i$ contains image with annotated bboxes of vehicles and a task description prompt, and $y_i$ contains the GT SIG (content of a JSON file, *e.g.* `vehicles:{black truck 1:[5,3]}; traffic_signs:{sign 1:[3,5]}...`) and SRP derived from GT SIG. Our main insight is to let the MLLM learn the corresponding spatial relation between object position in image and SIG and we demonstrate SIG's strong generalization ability with experiments in Sec. 4.3.

## 3.3 Human-Like VSI with Grid

In AD scenario, human decision-making is guided by selective attention: drivers focus on a subset of objects rather than all of them in the scene. We incorporate this human-like bias by integrating gaze or saliency predictions [93, 98] into SIG. Let $p_i = (u_i, v_i, 1)^\top$ denote an image pixel and $w_i = (X_i, Y_i, 1)^\top$ denote corresponding grid cell on SIG. We can calculate the the homographic matrix $H = [h_{ij}] \in \mathbb{R}^{3 \times 3}$ by solving the equations of $X_i$ and $Y_i$ as

$$X_i = \frac{h_{11}u_i + h_{12}v_i + h_{13}}{h_{31}u_i + h_{32}v_i + h_{33}}, \quad Y_i = \frac{h_{21}u_i + h_{22}v_i + h_{23}}{h_{31}u_i + h_{32}v_i + h_{33}} \qquad (5)$$

Table 2: **Quantitative comparison of different MLLMs on `SIGBench` dataset for general VSI tasks.** P, R, F1, and AssA means precision, recall, F1-score and Association Accuracy, respectively. S and WS means similarity ($\gamma, \beta = 1$) and weighted similarity ($\gamma \neq \beta$) in Eq. (3). Acc means accuracy. Dark blue and light blue indicates the best and the second best result among all models.

| Type | Models | MLSM | | | | SRGS | | SRD (Directional) | | | SRD (Proximal) | | |
|---|---|---|---|---|---|---|---|---|---|---|---|---|---|
| | | P↑ | R↑ | F1↑ | AssA↑ | S↑ | WS↑ | MAE↓ | MSE↓ | Acc↑ | MAE↓ | MSE↓ | Acc↑ |
| | Human-Level | 0.918 | 0.932 | 0.938 | 0.869 | 0.897 | 0.901 | 0.568 | 2.372 | 0.753 | 0.720 | 2.508 | 0.760 |
| Proprietary | Claude-3.5-Haiku | 0.415 | 0.371 | 0.373 | 0.243 | 0.225 | 0.203 | 2.179 | 6.247 | 0.103 | 0.701 | 0.960 | 0.416 |
| | Claude-3.7-Sonnet | 0.511 | 0.427 | 0.450 | 0.306 | 0.299 | 0.299 | 2.285 | 6.712 | 0.092 | 0.691 | 0.928 | 0.420 |
| | Gemini-1.5-Pro | 0.471 | 0.470 | 0.454 | 0.309 | 0.282 | 0.272 | 2.073 | 5.899 | 0.126 | 1.241 | 2.720 | 0.298 |
| | Gemini-2.0-Flash | 0.472 | 0.472 | 0.456 | 0.311 | 0.283 | 0.270 | 2.344 | 6.923 | 0.083 | 1.092 | 1.835 | 0.246 |
| | Gemini-2.5-Pro | 0.473 | 0.586 | 0.507 | 0.355 | 0.232 | 0.226 | 1.316 | 2.937 | 0.254 | 0.790 | 1.225 | 0.398 |
| | GPT-4o-mini | 0.167 | 0.156 | 0.154 | 0.087 | 0.054 | 0.019 | 2.478 | 7.451 | 0.061 | 0.922 | 1.442 | 0.312 |
| | GPT-4o | 0.507 | 0.441 | 0.458 | 0.315 | 0.337 | 0.331 | 1.942 | 5.309 | 0.144 | 0.949 | 1.563 | 0.313 |
| Open-source | InternVL3-9B | 0.290 | 0.302 | 0.284 | 0.174 | 0.133 | 0.086 | 2.019 | 5.804 | 0.130 | 0.942 | 1.524 | 0.314 |
| | InternVL3-14B | 0.432 | 0.374 | 0.391 | 0.258 | 0.254 | 0.234 | 2.305 | 6.850 | 0.070 | 1.139 | 1.957 | 0.272 |
| | InternVL2.5-26B | 0.386 | 0.383 | 0.369 | 0.239 | 0.220 | 0.189 | 2.006 | 5.752 | 0.135 | 0.940 | 1.470 | 0.310 |
| | Qwen-VL-2.5-7B | 0.251 | 0.306 | 0.258 | 0.156 | 0.102 | 0.066 | 2.426 | 7.413 | 0.066 | 1.279 | 2.574 | 0.215 |
| | Qwen-VL-2.5-32B | 0.427 | 0.350 | 0.375 | 0.245 | 0.248 | 0.223 | 2.080 | 6.077 | 0.113 | 1.650 | 3.732 | 0.128 |

Figure 5: (a) is the annotation pipeline of `SIGBench` and (b) illustrates the SIGC and SRPF tasks in `SIGBench`.

using singular value decomposition (SVD). We then project and normalize the raw attention map $A_{\text{Image}}$ with image size into the SIG attention map $A_{\text{SIG}}$ using

$$A_{\text{SIG}}(i,j) = \frac{A_{\text{Image}}\big(\lfloor x_{ij} \rfloor, \lfloor y_{ij} \rfloor\big) - \min_{u,v} A_{\text{Image}}(u,v)}{\max_{u,v} A_{\text{Image}}(u,v) - \min_{u,v} A_{\text{Image}}(u,v)}, \text{s.t.} [x_{ij}, y_{ij}, 1]^\top = H^{-1}[i,j,1]^\top \quad (6)$$

Thus, we can combine $A_{\text{SIG}}$ and SIG to create SIG with attention weight (*a.k.a.* human-like SIG).

**Human-Like Spatial Relation Graph Similarity.** To weight graph edit distance by human-like perceptual importance, we scale each node-edit cost by its corresponding attention weight from $A_{\text{SIG}}$. Edge-edit costs are similarly weighted by the average attention weight of the two incident nodes. This yields an attention-aware SRGS that penalizes errors on highly attended objects more severely.

**Human-Like Semantic Relational Distance.** To apply human gaze attention as a weight for the calculation of human-like SRD, we multiply the SRD between predicted and GT prepositions by the mean attention of the two referenced objects. For instance, if the GT relation is *black car 1 is {at the back left of} white car 2*, the predicted relation is *{at the front of}*, and the attention weights for black car 1 and white car 2 are 3 and 4. The semantic directional distance in this example is 3 (shown in Fig. 4) and the human-like SRD for this example can be computed as $3 \times \frac{3+4}{2} = 10.5$.

## 4 Experiment

### 4.1 SIG Benchmark

**Benchmark Overview.** We introduce `SIGBench`, a benchmark for quantifying both grid-based and human-like VSI in MLLMs within AD scenario. `SIGBench` comprises 1,423 frames, each annotated

with (i) SIG and human-like SIG, (ii) SRP and human-like SRP and (iii) gaze attention map in image size. The annotation pipeline is shown in Fig. 5 (a). Let $f$ denotes the camera focal length. We create attention map by firstly estimating a circular attention radius using

$$r = \min(r_w, r_h) \quad \text{s.t.} \quad r_w = \frac{f \cdot I_w}{s_w} \cdot \tan\left(\frac{\text{fov}_w}{2}\right), \quad r_h = \frac{f \cdot I_h}{s_h} \cdot \tan\left(\frac{\text{fov}_h}{2}\right) \qquad (7)$$

where $I_w, I_h$ are image width and height in pixels, $s_w, s_h$ are the corresponding sensor dimensions in millimeters, and $\text{fov}_w$ and $\text{fov}_h$ are the human horizontal and vertical field of view. Then we accumulate attention map from 6 consecutive frames to create GT attention map. For annotation of SIG, we annotate vehicles inside the bounding box using "color+type+order" label (*e.g.* black car 1), where order runs leftmost to rightmost in the image. We also annotate traffic signs/lights that are clearly visible using "type+order" (*e.g.* sign 1, light 1). Detailed annotation pipeline and data distribution of SIGBench and SIGBench-tiny regarding to number of objects in each sample can be found in Appendix B. In general, SIGBench contains two main task clusters: grid-based VSI tasks: spatial intelligence grid creation (SIGC) and spatial relation paragraph filling (SRPF) and human-like VSI tasks: human-like SIGC and SRPF, and gaze prediction.

Table 3: **Quantitative comparison of 3-shot ICL for general VSI tasks on SIGBench-tiny.** Z-S means zero-shot, ICL-MC meaning ICL using multiple-choice VQA and ICL-SIG meaning ICL using SIG. light red indicates the results that is worse than zero-shot after applying ICL on GPT-4o and Gemini-2.5-Pro.

| Models | Type | MLSM | | | | SRGS | | SRD (Directional) | | | SRD (Proximal) | | | MC (Acc) | |
|---|---|---|---|---|---|---|---|---|---|---|---|---|---|---|---|
| | | P↑ | R↑ | F1↑ | AssA↑ | S↑ | WS↑ | MAE↓ | MSE↓ | Acc↑ | MAE↓ | MSE↓ | Acc↑ | Dir.↑ | Prox.↑ |
| GPT-4o | Z-S | 0.522 | 0.432 | 0.462 | 0.316 | 0.327 | 0.321 | 1.792 | 4.827 | 0.186 | 0.858 | 1.324 | 0.346 | 0.056 | 0.292 |
| | ICL-MC | 0.545 | 0.431 | 0.468 | 0.320 | 0.324 | 0.323 | 1.600 | 4.106 | 0.218 | 0.920 | 1.621 | 0.365 | 0.144 | 0.337 |
| | ICL-SIG | 0.592 | 0.438 | 0.479 | 0.328 | 0.337 | 0.357 | 1.593 | 4.094 | 0.220 | 0.775 | 1.271 | 0.436 | 0.172 | 0.309 |
| Gemini-2.5-Pro | Z-S | 0.464 | 0.570 | 0.496 | 0.345 | 0.224 | 0.210 | 1.151 | 2.426 | 0.295 | 0.721 | 1.100 | 0.439 | 0.247 | 0.413 |
| | ICL-MC | 0.477 | 0.617 | 0.524 | 0.366 | 0.185 | 0.187 | 1.174 | 2.667 | 0.325 | 0.845 | 1.387 | 0.384 | 0.172 | 0.348 |
| | ICL-SIG | 0.556 | 0.608 | 0.565 | 0.406 | 0.305 | 0.307 | 1.126 | 2.396 | 0.316 | 0.578 | 0.729 | 0.493 | 0.305 | 0.447 |

(a) Results of Grid-based VSI Tasks    (b) ICL using GPT-4o    (c) ICL using Gemini-2.5-Pro

Figure 6: **Visualization of SIG-empowered VSI results.** SRD-Dir and SRD-Prox denotes the Acc in SRD (Directional) and SRF (Proximal). (a) demonstrate the performance of human and different models in grid-based tasks on SIGBench. Even for leading MLLMs, there is a substantial gap compared to human performance in VSI. (b) and (c) denotes the ICL results of GPT-4o and Gemini-2.5-pro on SIGBench-tiny. ICL-SIG outperforms the zero-shot baseline in all VSI metrics and delivers more general improvements than ICL-MC.

**Grid-based VSI Tasks.** For SIGC task, the MLLM is prompted to produce a $10\times10$ SIG in which every vehicle with bbox, traffic sign, traffic light, and the self (ego-vehicle) is placed according to its true world-coordinate position into a JSON file. The output SIG will be compared with corresponding GT SIG using using MLSM, SRGS. One example is shown in Fig. 5 (b) left. For SRPF task, it supplies each model with fully formed sentences that omit only the prepositional phrase between two object mentions. The model's objective is to choose the correct preposition in context. We target two complementary relation types: directional and proximal relations. For each image, we present two short paragraphs: one composed of blank slots for directional prepositions and one for proximal prepositions. One example is shown in Fig. 5 (b) right.

**Human-Like VSI with Grid Tasks.** For gaze prediction task, it evaluates an MLLM's ability to predict the human gaze attention map for frame $i$ based on attention map from frames $i-5$ to $i-1$ as human gaze always follow a spatial-temporal format. We want to measure how well the model anticipates where a driver would pay attention to given attention maps from previous frames.

Table 4: Average per-frame runtime analysis of metrics on AMD Ryzen 5900X CPU with varying object numbers. MLSM, SRGS, and SRD can be executed within sub-millisecond latency, satisfying real-time constraint.

| Metrics | Number of Objects | | | | | | | | | |
|---|---|---|---|---|---|---|---|---|---|---|
| | 3 | 5 | 7 | 9 | 11 | 13 | 15 | 17 | 19 | 22 |
| MLSM [s] | 0.0001 | 0.0002 | 0.0002 | 0.0002 | 0.0002 | 0.0002 | 0.0002 | 0.0003 | 0.0003 | 0.0004 |
| SRGS [s] | 0.0001 | 0.0002 | 0.0003 | 0.0004 | 0.0004 | 0.0005 | 0.0006 | 0.0006 | 0.0008 | 0.0009 |
| SRD [s] | <0.0001 | <0.0001 | <0.0001 | <0.0001 | <0.0001 | <0.0001 | <0.0001 | <0.0001 | <0.0001 | <0.0001 |

For human-like SIGC and SRPF tasks, they are similar as grid-based VSI SIGC and SRPF, but incorporate human gaze attention into the evaluation. Each object is assigned an attention weight from SIG attention map $A_{\mathrm{SIG}}^i$ for frame $i$, reflecting its relative importance to a human observer. These attention-weighted tasks are used to reveal whether an MLLM can prioritize spatial relations between objects according to their importance to current scene in a human-like manner.

## 4.2 Model Selection

**Models.** We evaluate several top-tier MLLMs on `SIGBench` mainly from five modal families: 1) open-source models such as InternVL [49] and Qwen-VL [44]; 2) Proprietary models including OpenAI GPT [43], Google Gemini [45] and Anthropic Claude [48]. The specific models in each model family with their complete names we used during experiment are listed in detail in Tab. 2.

**Evaluation Metrics.** For gaze prediction tasks, we follow the widely used metrics in gaze/saliency prediction task such as Person's Correlation Coefficient (PCC), KL-Divergence and Information Gain (IG) [93, 98]. For grid-based SIGC and SRPF, we use MLSM, SRGS and SRD mentioned in Sec. 3.1. For human-like SIGC and SRPF, we use human-like SRGS and SRD mentioned in Sec. 3.3.

## 4.3 Results and Analysis for Grid-based VSI on SIGBench

**Zero-shot Inference on `SIGBench`, showing in Tab. 2.** The performance of several MLLMs in grid-based SIGC and SRPF using zero-shot inference are shown in Tab. 2. In general, Gemini-2.5-Pro achieves the best performance in both MLSM and SRD (Directional), illustrating its strong ability in spatial understanding of object position and direction. GPT-4o achieves the best in SRGS and the second best in MLSM and SRD, revealing its strong capability in figuring spatial-relation between objects. Besides, Claude-3.7-Sonnet demonstrate strong capability in understanding proximal distance between objects in text and second best in SRGS. By analyzing failure cases, we observe that small or peripheral objects are often missed or mislocalized and substantial overlap (high IoU) between objects exacerbates this by causing identity conflation and incorrect grid-cell placement.

**SIG-based ICL with Random Sample Selection, showing in Tab. 3.** To evaluate SIG's advantage over conventional VQA-style representation for VSI, we conduct ICL on GPT-4o and Gemini-2.5-Pro, whose performance are outstanding among all models in Tab. 2. We randomly select 90 samples from `SIGBench` as `SIGBench-tiny` and generate 4–8 multiple-choice questions targeting directional and proximal relations, mimicking existing VQA benchmarks for VSI for each image. In addition, we randomly selected 3 images (outside `SIGBench-tiny`) and annotated full VQA pairs covering every object in their ground-truth SIGs for training. we conduct a 3-shot ICL using SIG (ICL-SIG) and VQA (ICL-MC) annotations as the input representation, respectively and evaluate on `SIGBench-tiny` using our proposed SIGC and SRPF metrics alongside the accuracy of annotated VQA tasks (MC).

The key findings are: 1) the model's VSI consistently improves with ICL-SIG. Across both models, ICL-SIG improves nearly every metric relative to zero-shot baselines; 2) Even using randomly sampled images as context examples without sophisticated sampling strategy, ICL-SIG generally improves VSI over ICL-MC, especially for MLSM, SRGS and SRD (Proximal); 3) Compared to ICL-SIG, the improvements brought by ICL-MC is unstable, which result in lower performance than zero-shot in metrics such as SRGS-S in both models. To demonstrate the potential variability of SIG-based ICL with difference choice of data samples, we conduct further ablation studies in Appendix B.4. These findings highlights SIG's superior fidelity in encoding VSI compared to traditional VQA and its potential as a new representation schema for improving VSI in MLLMs.

**Empirical Runtime Analysis of Evaluation Metrics** Runtime efficiency is crucial for real-time applications like AD. We report the empirical per-frame runtimes of our proposed evaluation metrics on `SIGBench` under varying object numbers, as summarized in Tab. 4.

Table 5: **Results of zero-shot and 3-shot ICL on SIG-COCO and SIG-ARKitScenes for general VSI tasks using GPT-4o.** Z-S means zero-shot, ICL-SIG meaning ICL using SIG.

| Benchmark | Type | MLSM | | | | SRGS | | SRD (Directional) | | | SRD (Proximal) | | |
|---|---|---|---|---|---|---|---|---|---|---|---|---|---|
| | | P↑ | R↑ | F1↑ | AssA↑ | S↑ | WS↑ | MAE↓ | MSE↓ | Acc↑ | MAE↓ | MSE↓ | Acc↑ |
| SIG-COCO | Z-S | 0.758 | 0.826 | 0.771 | 0.652 | 0.574 | 0.607 | 1.303 | 3.403 | 0.407 | 0.893 | 1.160 | 0.230 |
| | ICL-SIG | 0.860 | 0.840 | 0.849 | 0.749 | 0.711 | 0.746 | 0.713 | 1.827 | 0.640 | 0.357 | 0.360 | 0.643 |
| SIG-ARKitScenes | Z-S | 0.714 | 0.756 | 0.727 | 0.581 | 0.619 | 0.597 | 1.267 | 3.467 | 0.433 | 0.900 | 1.167 | 0.233 |
| | ICL-SIG | 0.809 | 0.850 | 0.823 | 0.719 | 0.729 | 0.737 | 0.467 | 0.533 | 0.567 | 0.300 | 0.500 | 0.800 |

Table 6: **Quantitative comparison of different MLLMs on `SIGBench` dataset for human-like visual-spatial intelligence tasks.** H means human-like and KL-D means KL-Divergence.

| Type | Models | Gaze Prediction | | | H-SRGS | | H-SRD (Directional) | | | H-SRD (Proximal) | | |
|---|---|---|---|---|---|---|---|---|---|---|---|---|
| | | PCC↑ | KL-D↓ | IG↑ | S↑ | WS↑ | MAE↓ | MSE↓ | Acc↑ | MAE↓ | MSE↓ | Acc↑ |
| Proprietary | Claude-3.5-Haiku | 0.798 | 0.520 | 0.364 | 0.106 | 0.072 | 1.265 | 3.291 | 0.111 | 0.374 | 0.488 | 0.388 |
| | Claude-3.7-Sonnet | 0.772 | 0.344 | 0.623 | 0.157 | 0.127 | 1.346 | 3.607 | 0.096 | 0.350 | 0.428 | 0.398 |
| | Gemini-1.5-Pro | 0.914 | 0.125 | 0.946 | 0.161 | 0.123 | 1.209 | 3.081 | 0.119 | 0.713 | 1.430 | 0.259 |
| | Gemini-2.0-Flash | 0.861 | 0.231 | 0.803 | 0.160 | 0.121 | 1.439 | 3.956 | 0.084 | 0.631 | 0.969 | 0.232 |
| | Gemini-2.5-Pro | 0.868 | 0.314 | 0.637 | 0.148 | 0.118 | 0.784 | 1.621 | 0.222 | 0.450 | 0.659 | 0.349 |
| | GPT-4o-mini | 0.838 | 0.514 | 0.504 | 0.010 | 0.003 | 1.509 | 4.165 | 0.064 | 0.484 | 0.693 | 0.324 |
| | GPT-4o | 0.673 | 0.506 | 0.406 | 0.197 | 0.163 | 1.146 | 2.912 | 0.145 | 0.557 | 0.870 | 0.287 |
| Open-source | InternVL3-9B | 0.427 | 2.932 | -2.652 | 0.059 | 0.030 | 1.303 | 3.450 | 0.105 | 0.525 | 0.669 | 0.250 |
| | InternVL3-14B | 0.859 | 0.860 | -0.073 | 0.131 | 0.099 | 1.338 | 3.585 | 0.093 | 0.780 | 1.448 | 0.182 |
| | InternVL2.5-26B | 0.951 | 0.090 | 0.988 | 0.120 | 0.082 | 1.247 | 3.225 | 0.136 | 0.539 | 0.669 | 0.254 |
| | Qwen-VL-2.5-7B | 0.950 | 0.068 | 1.028 | 0.038 | 0.018 | 1.492 | 4.200 | 0.081 | 0.958 | 2.033 | 0.152 |
| | Qwen-VL-2.5-32B | 0.664 | 1.373 | -0.908 | 0.123 | 0.087 | 1.302 | 3.433 | 0.102 | 1.231 | 2.875 | 0.081 |

**Cross-Domain Generalizability.** Although our initial motivation arises from AD, SIG as a data representation is fundamentally domain-agnostic and can be applied wherever a fixed ontology of object types exists. For demonstration, we construct two proof-of-concept benchmarks based on subsets from MS COCO [107] and ARKitScenes [108], which we denote as SIG-COCO and SIG-ARKitScenes, respectively. We conduct both zero-shot inference and ICL experiments with GPT-4o on these benchmarks, with results reported in Tab. 5. ICL-SIG consistently outperforms zero-shot inference across both benchmarks, suggesting SIG generalizes effectively beyond AD domain.

## 4.4 Results and Analysis for Human-Like VSI with Grid on SIGBench

The results for gaze prediction, human-like SIGC and SRPF are shown in Tab. 6. Surprisingly, InternVL2.5-26B and Qwen-VL-2.5-7B achieves the best performance in gaze prediction task. After looking at the their output, we found that different from other models that apply operations such as gaussian blur, edge detection after taking the average of five previous attention map, these two models only take the average (details in Appendix C.3). Model rankings under human-SRGS and human-SRD closely mirror the grid-based VSI results (Tab. 2), indicating that incorporating attention weights doesn't alter relative performance between MLLMs. This shows that current MLLMs still struggle to prioritize scene objects with human-like selectivity in AD settings.

## 5 Conclusion and Discussion

We propose a novel representation for visual-spatial intelligence (VSI) called *spatial intelligence grid* (SIG) and introduce a suite of graph-based evaluation metrics that leverage its structured topology to enable more precise and general VSI assessment. Through experiments on different MLLMs, we demonstrate that SIG-based few-shot in-context learning consistently delivers larger, more stable and more comprehensive VSI enhancement than using traditional VQA-style prompts solely, underscoring SIG's superior capacity to encode complex spatial relations. Based on SIG, we create `SIGBench`, a benchmark with image-SIG/human gaze attention pairs, which is designed to evaluate both grid-based machine VSI and human-like attention-driven spatial reasoning under our proposed metrics. Taken together, these contributions offer a principled data schema and a practical yardstick for VSI. Despite these advances, our study has two limitations remaining for future works: (i) `SIGBench` currently targets single-frame settings and therefore does not assess tracking or dynamic object–object relations that require temporal context; and (ii) while SIG proves effective for in-context learning, SIG-based fine-tuning and reinforcement learning with human feedback remain unexplored.

# 6 Acknowledgement

The author team would like to share the sincere thank to Wei Zhang from U.S. Department of Transportation (USDOT) for providing the valuable U.S. Federal Highway Administration driving dataset, based on which we construct our proposed benchmark. The author team also appreciate the valuable suggestions from Junyue Jiang, Linshen Liu and Yibo Zhao during the discussion.

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

# Towards Physics-informed Spatial Intelligence with Human Priors: An Autonomous Driving Pilot Study

## Supplementary Material

## Contents

# A  Evaluation Metrics

## A.1  Multi-level Spatial Matching

Let $\alpha \in [1, \ldots, n]$ denote $n$ thresholds based on euclidean distance, we can calculate

$$\mathrm{P}_\alpha = \frac{TP_\alpha}{TP_\alpha + FP_\alpha}, \mathrm{R}_\alpha = \frac{TP_\alpha}{TP_\alpha + FN_\alpha}, \mathrm{AssA}_\alpha = \frac{TP_\alpha}{TP_\alpha + FP_{\alpha_k} + FN_{\alpha_k}}. \quad (8)$$

Then, we normalize each of them through $n$ thresholds

$$\mathrm{P} = \frac{1}{n} \sum_{\alpha=1}^{n} \mathrm{P}_\alpha, \;\; \mathrm{R} = \frac{1}{n} \sum_{\alpha=1}^{n} \mathrm{R}_\alpha, \;\; \mathrm{F1} = \frac{2\mathrm{PR}}{\mathrm{P} + \mathrm{R}}, \;\; \mathrm{AssA} = \frac{1}{n} \sum_{\alpha=1}^{n} \mathrm{AssA}_\alpha \quad (9)$$

## A.2  Spatial Relation Graph Similarity

For the calculation of spatial relation graph similarity, we have two main parts: the node edit distance $D_N(\mathcal{G}, \hat{\mathcal{G}})$ and edge edit distance $D_E(\mathcal{G}, \hat{\mathcal{G}})$. The node edit distance $D_N(\mathcal{G}, \hat{\mathcal{G}})$ includes substitution cost $\delta_{\mathrm{sub}}(v_i, \hat{v}_{i'})$, deletion cost $\delta_{\mathrm{del}}(v_i)$ and insertion cost $\delta_{\mathrm{ins}}(\hat{v}_j)$. Let $d^N_{v_i, v_{i'}}$ denotes the euclidean distance between the position of node $v_i$ and $v_{i'}$ on SIG and $\lambda_N$ as the total edit distance for the unmatch of node attributes (*e.g.* color, type and order). The detailed formulas for each cost can be shown as

$$\delta_{\mathrm{sub}}(v_i, \hat{v}_{i'}) = \begin{cases} d^N_{v_i, v_{i'}} + \lambda_N \, \mathbb{I}\big[\mathrm{attr}(v_i) \neq \mathrm{attr}(\hat{v}_{i'})\big], & \text{if } (v_i, \hat{v}_{i'}) \in \mathcal{M}, \\ 0, & \text{otherwise.} \end{cases} \quad (10)$$

$$\delta_{\mathrm{del}}(\hat{v}_i) = \begin{cases} \eta^N_{\mathrm{del}}, & \text{if } \hat{v}_i \notin \{\, v \mid (v, \hat{v}) \in \mathcal{M} \,\}, \\ 0, & \text{otherwise.} \end{cases} \quad (11)$$

$$\delta_{\mathrm{ins}}(v_j) = \begin{cases} \eta^N_{\mathrm{ins}}, & \text{if } v_j \notin \{\, v \mid (v, \hat{v}) \in \mathcal{M} \,\}, \\ 0, & \text{otherwise.} \end{cases} \quad (12)$$

$$(13)$$

where $\eta^N_{\mathrm{del}}, \eta^N_{\mathrm{ins}}$ are the cost for node insertion and deletion. For the edge edit distance $D_E(\mathcal{G}, \hat{\mathcal{G}})$, it includes edge substitution cost $\delta^E_{\mathrm{sub}}(e_i, \hat{e}_{i'})$, edge deletion cost $\delta^E_{\mathrm{del}}(\hat{e}_i)$, and edge insertion cost $\delta^E_{\mathrm{ins}}(e_j)$. Let $d^E_{e_i, e_{i'}}$ denotes the length between edge $e_i$ and $e_{i'}$ and $\lambda_E$ as the total edit distance for the unmatch of edge attributes (*e.g.* direction).

$$\delta^E_{\mathrm{sub}}(e_i, \hat{e}_{i'}) = \begin{cases} d^E_{e_i, e_{i'}} + \lambda_E \, \mathbb{I}\big[\mathrm{attr}(e_i) \neq \mathrm{attr}(\hat{e}_{i'})\big], & \text{if } (e_i, \hat{e}_{i'}) \in \mathcal{M}_\mathcal{E}, \\ 0, & \text{otherwise.} \end{cases} \quad (14)$$

$$\delta^E_{\mathrm{del}}(\hat{e}_i) = \begin{cases} \eta^E_{\mathrm{del}}, & \text{if } \hat{e}_i \notin \{\, e \mid (e, \hat{e}) \in \mathcal{M}_\mathcal{E} \,\}, \\ 0, & \text{otherwise.} \end{cases} \quad (15)$$

$$\delta^E_{\mathrm{ins}}(e_j) = \begin{cases} \eta^E_{\mathrm{ins}}, & \text{if } e_j \notin \{\, e \mid (e, \hat{e}) \in \mathcal{M}_\mathcal{E} \,\}, \\ 0, & \text{otherwise.} \end{cases} \quad (16)$$

where $\eta^E_{\mathrm{del}}, \eta^E_{\mathrm{ins}}$ are the cost for edge insertion and deletion and $\mathcal{M}_\mathcal{E}$ is the set of matched edges based on $\mathcal{M}$.

## A.3  Semantic Relational Distance

Let $\tilde{d}_i$ denote the signed distance (in steps) between the predicted and ground-truth preposition for the $i$-th example. We can calculate MAE and MSE by

$$MAE = \frac{1}{n} \sum_{i=1}^{n} |\tilde{d}_i| \quad MSE = \frac{1}{n} \sum_{i=1}^{n} |\tilde{d}_i|^2. \quad (17)$$

# B   SIG Benchmark

## B.1   Overview

We introduce `SIGBench`, a benchmark for quantifying both grid-based and human-like VSI in MLLMs within AD scenario. `SIGBench` comprises 1,423 frames, each annotated with (i) SIG and human-like SIG, (ii) SRP and human-like SRP and (iii) a corresponding gaze attention map in image size. All image sequences and raw gaze data are drawn from the U.S. Federal Highway Administration driving dataset, which includes 49 driving sessions ($\approx$ 65 minutes each at 25 FPS) recorded from 25 drivers, with gaze tracked by SmartEye Pro5. `SIGBench` contains two main parts: grid-based VSI, containing tasks such as spatial intelligence grid creation (SIGC) and spatial relation paragraph filling (SRPF) and human-like VSI, containing human-like SIGC and SRPF, and gaze prediction.

## B.2   Benchmark Annotation

We begin by filtering out frames with missing or erratic gaze data, retaining only those that have valid gaze points in the current frame and the previous five frames. For each selected frame, we estimate a circular attention radius $r$ based on

$$r = \min(r_w, r_h) \quad \text{s.t.} \quad r_w = \frac{f \cdot I_w}{s_w} \cdot \tan(\frac{\text{fov}_w}{2}), \quad r_h = \frac{f \cdot I_h}{s_h} \cdot \tan(\frac{\text{fov}_h}{2}) \qquad (18)$$

where $f$ is the camera focal length, $I_w, I_h$ are image width and height in pixels, $s_w, s_h$ are the corresponding sensor dimensions in millimeters, and $\text{fov}_w$ and $\text{fov}_h$ are the human horizontal and vertical field of view. We then build the per-frame attention map $A^i_{\text{Image}}$ by accumulating attention map from frame $i - 5$ to $i$. Next, we detect vehicles in each selected frame using Grounded-SAM-2 [109] with the prompts "vehicles" and "trucks" with thresholds including: confidence $\geq 0.28$, bounding box area $\geq 1900$ and text $\geq 0.25$. We then input the original image and vehicle bboxes into Gemini-2.0-Flash and let it output JSON type files containing SIG. The prompt we use are shown in Prompt. B.1. To correct spurious or missing labels, human annotators check and refine all annotated SIG. For vehicles, we only keep those with valid bounding boxes and categorize each as one of four types—truck, bus, car, or van—and assign a "color+type+order" label (*e.g.* black car 1), where order runs leftmost to rightmost in the image. All traffic signs and lights that are clearly visible (*i.e.* not too small or distant) are likewise annotated with "type+order" (*e.g.* sign 1, light 1).

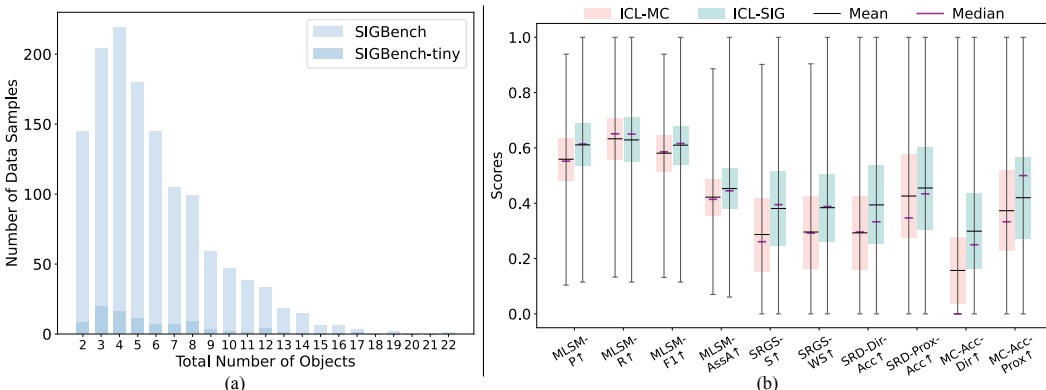

Figure 7: (a) Data distribution of `SIGBench` and `SIGBench`-tiny with object numbers. (b) Error bar statistics of multi-run ICL on `SIGBench`-tiny using Gemini-2.5-Pro with random data selection. The bars show the range of $\mu \pm \frac{1}{2}\sigma$ and whiskers indicate min and max of each metrics. Across metrics, ICL-SIG generally yields higher means/medians than ICL-MC.

## B.3   Models for Experiment

For experiment on `SIGBench`, the detailed models we used are listed here. For open-source models: InternVL family (InternVL-2.5-26B, InternVL-3-9B, InternVL-3-14B) [49], Qwen-VL family (Qwen-VL-2.5-7B, Qwen-VL-2.5-32B) [44], LLaMA family (LLaMA-3.2-11B-Vision-Instruct) [46] and DeepSeek-VL family (DeepSeek-VL, DeepSeek-VL-2-Tiny, DeepSeek-VL-2-Small) [47]. However, since LLaMA-3.2-11B-Vision-Instruct

and DeepSeek-VL models cannot correctly output JSON type file for SIG creation after several prompt engineering, we cannot record their performance. For proprietary models, we evaluate the performance of OpenAI GPT family (GPT-4o-mini, GPT-4o) [43], Google Gemini family (Gemini-1.5-Pro, Gemini-2.0-Flash, Gemini-2.5-Pro) [45] and Anthropic Claude family (Claude-3.7-Haiku, Claude-3.7-Sonnet) [48].

### B.4 Data Distribution and Ablation Study

In this section, we provide the detailed data distribution of `SIGBench` and `SIGBench`-tiny regarding object number shown in Fig. 7 (a). To demonstrate the potential variability of SIG-based ICL with difference choice of data samples, we conduct further experiments and provide the error bar statistics in Appendix B.4. These findings highlights SIG's superior fidelity in encoding VSI compared to traditional VQA and its potential as a new representation schema for improving VSI in MLLMs. To provide more insights about the impacts brought by different selection of data samples to the performance of SIG-based ICL, we conduct five independent runs with randomly selected data samples using Gemini-2.5-Pro and provide the error bar statistics shown in Fig. 7 (b). As shown in Fig. 7 (b), SIG-based ICL shows a consistent improvement than ICL using MC across almost all metrics. The results demonstrate that SIG encodes VSI with greater fidelity than traditional VQA, positioning it as a promising new representation schema to enhance VSI in MLLMs.

**Prompt B.1: Initial SIG Annotation using Gemini**

[Task Summary] The first image captures an outdoor driving scene, and the second image is the same scene with bounding boxes on certain vehicles. Please identify vehicles, traffic lanes, traffic signs, and traffic lights within the image, and understand the spatial arrangement of these entities. Specifically, assuming the bird's eye view of the scene is represented by a 10x10 grid, please estimate the center position of these entities within this grid. The output is expected to be a JSON file containing a dictionary.

[Task Details] <Overall> 1. The first image captures an outdoor driving scene, and the second image is the same scene but only with bounding boxes on certain vehicles. 2. Estimate the center positions of each instance within the first image, assuming the entire scene is represented by a 10x10 bird's eye view grid. 3. The entities to be estimated include: vehicles, traffic lanes, traffic signs, traffic lights, and the vehicle that capture the images (self). 4. Both the horizontal and vertical coordinates of the grid range from 0 to 9, so all estimated positions (*e.g.*, [x_1, y_1]) must fall within this range. 5. Estimated location of each instance should accurately reflect its real position in the scene, preserving the relative spatial relationships among all objects. 6. Please be aware of the front-backward or left-right relationship between instances, as there will be partial occlusion. 7. Since it is a bird's eye view grid, for all instances, more far away objects should be placed in the higher row number and the closer objects should be placed in the lower row number. <Vehicles> 1. Detect vehicles in the first image that are enclosed by bounding boxes in the second image only, and estimate the center positions of these enclosed vehicles within the grid. 2. The output is a key-value pair, which is expected to be exactly like: "vehicles": {"black car 1": [x_1, y_1], "gray truck 2": [x_2, y_2], ...}, where each vehicle instance is named by color, vehicle type, and order. 3. The color of vehicles are summarized into: gray, black, white, silver, blue, green, yellow, red, purple. Other colors need to be attributed to the one closest to it among these colors. 4. The type of vehicles here are summarized into four category: car(including suv), truck(including pickup truck), van, and bus. 5. The order of vehicles is decided exactly from left to right of each vehicle in the first image, the left most vehicle is 1, the second left most vehicle is 2, etc. <Traffic lanes> 1. Detect all traffic lanes of the same direction of the vehicle that captures the image in the first image, and estimate the lane position within the grid. One lane can be represented by multiple adjacent points as it is long. 2. The output is a key-value pair, which is expected to be exactly like: "traffic_lanes": {"lane 1": [[x_11, y_11], [x_12, y_12], ...], "lane 2": [[x_21, y_21], [x_22, y_22], ...], ...}, where each lane is named by order. 3. The order of lanes is decided exactly from left to right of each lane in the first image, the left most lane is 1, the second left most lane is 2, etc. <Traffic signs> 1. Detect all traffic signs in the first image, and estimate the center positions of these traffic signs. 2. The output is a key-value pair, which is expected to be exactly like: "traffic_signs": {"sign 1": [x_1, y_1], "sign 2": [x_2, y_2], ...}, where each sign is named by order. 3. The order of signs is decided exactly from left to right of each sign in the first image, the left most sign is 1, the second left most sign is 2, etc. 4. If multiple signs are mounted on the same horizontal pole, please treat them as separate instances. <Traffic lights> 1. Detect all traffic lights in the first image, and estimate the center positions of these traffic lights. 2. The output is a key-value pair, which is expected to be exactly like: "traffic_lights": {"light 1": [x_1, y_1], "light 2": [x_2, y_2], ...}, where each light is named by order. 3. The order of lights is decided exactly from left to right of each light in the first image, the left most light is 1, the second left most light is 2, etc. 4. If multiple traffic lights are mounted on the same horizontal pole, please treat them as one single light and use the midpoint between them as the center. <Self> 1. Estimate the center location of the vehicle that captures the image. 2. The output is expected to be exactly like: "self": [x, 0] 3. The vehicle capturing the image should be placed in the point with raw index 0 and with column index depends on different images.

[Output] 1. Combine the key-value pairs of vehicles, traffic lanes, traffic signs and traffic lights into one dictionary, as final output. 2. The final output dictionary is expected to be exactly like: {
`"vehicles": {"black car 1": [x_1, y_1], "gray truck 2": [x_2, y_2], ...},`
`"traffic_lanes": {"lane 1": [[x_11, y_11], [x_12, y_12], ...], "lane 2":`
`[[x_21, y_21], [x_22, y_22], ...], ...}, "traffic_signs": {"sign 1": [x_1,`
`y_1], "sign 2": [x_2, y_2], ...}, "traffic_lights": {"light 1": [x_1, y_1],`
`"light 2": [x_2, y_2], ...}, "self": [x, 0] }` 3. Only output the final dictionary as a JSON file.

# C   SIGBench Task Examples

In this section, we provide the prompt we used for tasks including spatial intelligence grid creation (SIGC) (Sec. C.1), Spatial Relation Paragraph Filling (SRPF) (Sec. C.2), Gaze prediction (Sec. C.3), human-like SIGC (Sec. C.4) and human-like SRPF (Sec. C.5). Since different models need slight modification to correctly output the format we want, here we only provide some of them for illustration.

## C.1   Spatial Intelligence Grid Creation

---

**Prompt C.1: SIGC Example General**

Q: [SIGC task prompt] + [Original Image] + [Original Image with bbox of vehicles].
A:        {"vehicles":{"white car 1":[5,3],"gray truck 2":[4,2],"yellow
truck 3":[7,1]},"traffic_lanes":{"lane 1":[[3,0],[3,2]],"lane
2":[[4,1],[4,7]],"lane 3":[[5,0],[5,1]]},"traffic_signs":{"sign
1":[3,3],"sign 2":[8,1]},"traffic_lights":{"light 1":[6,1],"light
2":[5,2]},"self":[4,0]}.

---

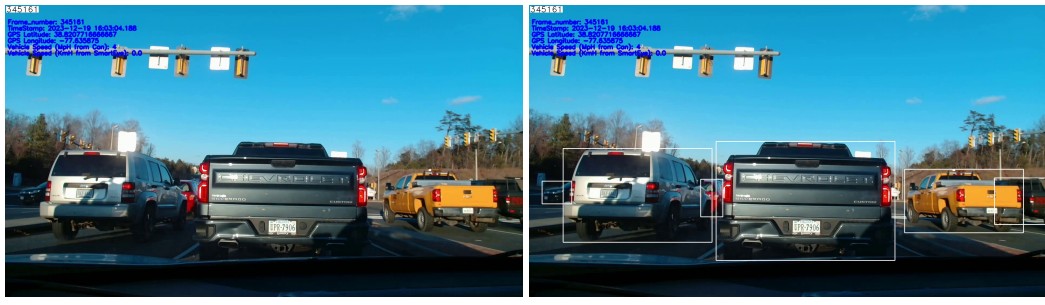

(a) Original image                              (b) Image with bbox of vehicles

Figure 8: Example of input images for SIGC task

For SIGC, the general structure of this task is shown as Prompt. C.1. The detailed prompt we input is shown in Prompt. C.2. The original image and image with bbox of vehicles are shown in Fig. 8a and Fig. 8b, respectively. The visualization of predicted SIG output by GPT-4o for this image is shown in Fig. 9a and GT SIG is shown in Fig. 9b.

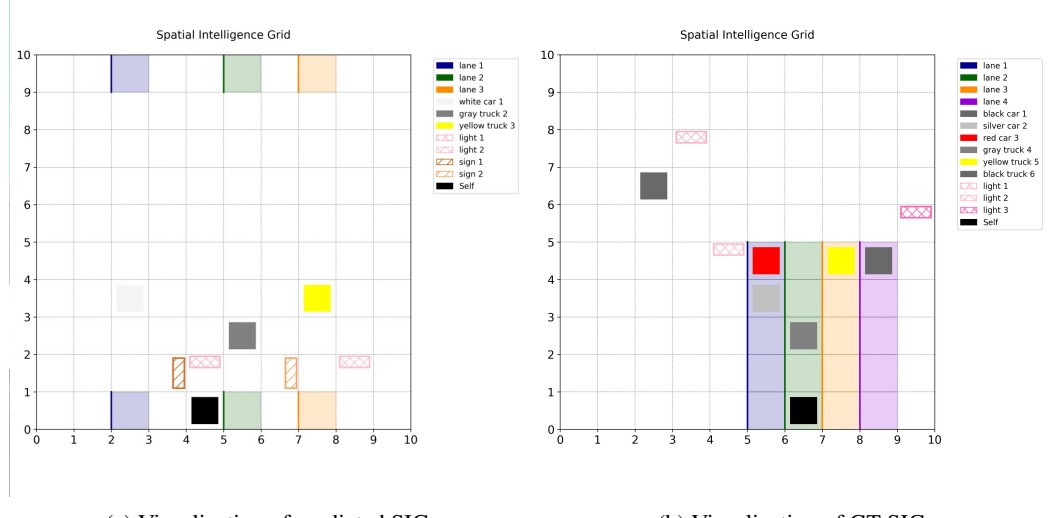

(a) Visualization of predicted SIG                    (b) Visualization of GT SIG

Figure 9: Visualization of predicted SIG and GT SIG

**Prompt C.2: SIGC Prompt Example**

[Task Summary] The first image captures an outdoor driving scene, and the second image is the same scene with bounding boxes on certain vehicles. Please identify vehicles, traffic lanes, traffic signs, and traffic lights within the image, and understand the spatial arrangement of these entities. Specifically, assuming the bird's eye view of the scene is represented by a 10*10 grid, please estimate the center position of these entities within this grid. The output is expected to be a JSON file containing a dictionary.

[Task Details] <Overall> 1. The first image captures an outdoor driving scene, and the second image is the same scene but only with bounding boxes on certain vehicles. 2. Estimate the center positions of each instance within the first image, assuming the entire scene is represented by a 10*10 bird's eye view grid. 3. The entities to be estimated include: vehicles, traffic lanes, traffic signs, traffic lights, and the vehicle that capture the images (self). 4. Both the horizontal and vertical coordinates of the grid range from 0 to 9, so all estimated positions (e.g., [x_1, y_1]) must fall within this range. 5. Estimated location of each instance should accurately reflect its real position in the scene, preserving the relative spatial relationships among all objects. 6. Please be aware of the front-backward or left-right relationship between instances, as there will be partial occlusion. 7. Since it is a bird's eye view grid, for all instances, more far away objects should be placed in the higher row number and the closer objects should be placed in the lower row number. <Vehicles> 1. Detect vehicles in the first image that are enclosed by bounding boxes in the second image only, and estimate the center positions of these enclosed vehicles within the grid. 2. The output is a key-value pair, which is expected to be exactly like: "vehicles": {"black car 1": [x_1, y_1], "gray truck 2": [x_2, y_2], "yellow bus 3": [x_3, y_3], "blue van 4": [x_4, y_4], ...}, where each vehicle instance is named by color, vehicle type, and order. 3. The color of vehicles are summarized into: gray, black, white, silver, blue, green, yellow, red, purple. Other colors need to be attributed to the one closest to it among these colors. 4. The type of vehicles must be classified into exactly four categories: "car" (which include sedans, hatchbacks, and suvs), "truck" (including pickup trucks), "van", and "bus". Do not use "suv" or any other subcategory names directly as a type. Instead, map them into one of the four allowed categories. 5. The order of vehicles must be assigned globally, based strictly on their horizontal position in the first image. The left-most vehicle in the image must be numbered "1", the second left-most vehicle must be numbered "2", and so on. This numbering should apply across all vehicles, without restarting for different colors or types. 6. Vehicles usually travel within traffic lanes, so the estimated position of a vehicle moving in the same direction as the "self" vehicle should typically fall within the coordinates of a corresponding lane. This does not apply to vehicles traveling in the other directions or that are stationary. <Traffic lanes> 1. Detect all traffic lanes in the same driving direction as the vehicle capturing the image, and estimate the lane position within the grid. One lane can be represented by multiple adjacent points as it is long. 2. The output is a key-value pair, which is expected to be exactly like: "traffic_lanes": {"lane 1": [[x_11, y_11], [x_12, y_12], ...], "lane 2": [[x_21, y_21], [x_22, y_22], ...], ...}, where each lane is named by order. 3. The order of traffic lanes is based strictly on their horizontal position in the first image. The left-most lane in the image must be numbered "1", the second left-most lane must be numbered "2", and so on. 4. Each lane is typically straight, so the horizontal coordinates of its points should usually be the same, forming a vertical line in the grid, unless the lane is clearly turning, merging, or splitting. 5. Additionally, adjacent lanes are typically close together, so their horizontal coordinate values of adjacent lanes should usually differ by only 1. <Traffic signs> 1. Detect all traffic signs in the first image, and estimate the center positions of these traffic signs. 2. The output is a key-value pair, which is expected to be exactly like: "traffic_signs": {"sign 1": [x_1, y_1], "sign 2": [x_2, y_2], ...}, where each sign is named by order. 3. The order of traffic signs is based strictly on their horizontal position in the first image. The left-most sign in the image must be numbered "1", the second left-most sign must be numbered "2", and so on. 4. If multiple signs are mounted on the same horizontal pole, please treat them as separate instances. <Traffic lights> 1. Detect all traffic lights in the first image, and estimate the center positions of these traffic lights. 2. The output is a key-value pair, which is expected to be exactly like: "traffic_lights": {"light 1": [x_1, y_1], "light 2": [x_2, y_2], ...}, where each light is named by order. 3. The order of traffic lights is based strictly on their horizontal position in the first image. The left-most light in the image must be numbered "1", the second left-most light must be numbered "2", and so on. 4. If multiple traffic lights are mounted on the same horizontal pole, please treat them as one single light and use the midpoint between them as the center. <Self> 1. Estimate the center location of the vehicle that captures the image. 2. The output is expected to be exactly like: '"self": [x, 0]' 3. The vehicle capturing the image should be placed in the point with raw index 0 and with column index depends on different images.

[Output] 1. Combine the key-value pairs of vehicles, traffic lanes, traffic signs and traffic lights into one dictionary, as final output. 2. The final output dictionary is expected to be exactly like: { "vehicles": {"black car 1": [x_1, y_1], "gray truck 2": [x_2, y_2], "yellow bus 3": [x_3, y_3], "blue van 4": [x_4, y_4], ...}, "traffic_lanes": {"lane 1": [[x_11, y_11], [x_12, y_12], ...], "lane 2": [[x_21, y_21], [x_22, y_22], ...], ...}, "traffic_signs": {"sign 1": [x_1, y_1], "sign 2": [x_2, y_2], ...}, "traffic_lights": {"light 1": [x_1, y_1], "light 2": [x_2, y_2], ...}, "self": [x, 0] } 3. Please output the final dictionary in pure JSON format, without any additional text or explanation before or after. 4. All keys and string values in the output dictionary must be in lowercase letters only.

## C.2 Spatial Relation Paragraph Filling

---

**Prompt C.3: SRPF Example General**

Q: [SRPF task prompt] + [Original Image] + [Original Image with bbox of vehicles] + [Directional Relation Paragraph Template] + [Proximal Relation Paragraph Template].
A: {"answers_directional":[2,2,4,4,4,4,4,4,2,4,4,4,4,4,4,4,4,4,4,4,4,4,6,4,4,4,4,4,4,4,4,4,4,4,4],
"answers_proximal":[0,0,1,1,2,2,2,0,1,1,1,2,2,2,1,1,1,2,2,2,1,1,2,2,2,1,1,2,2,2,2,2,2,2]}.

---

For SRPF, the general structure of this task is shown as Prompt. C.3. The output answer is corresponding to the index of DIRECTIONAL_LABELS = ["at the back of", "at the back left of", "at the left of", "at the front left of", "at the front of", "at the front right of", "at the right of", "at the back right of"] and PROXIMAL_LABELS = ["adjacent to", "close to", "at a distance", "far from", "far away from"]. The detailed prompt we input is shown in Prompt. C.4. The original image and image with bbox of vehicles are shown in Fig. 8a and Fig. 8b, respectively.

---

**Prompt C.4: SRPF Prompt Example**

[Input Description] You will be provided with two images and two incomplete text paragraphs. 1. The first image shows a real-world driving scene. 2. The second image is the same scene with bounding boxes drawn on several vehicles. 3. The first paragraph is a Directional Template that describes the directional relationships between certain vehicles, traffic signs, or traffic lights, using [directional preposition] as placeholders. Example: "Black car 1 is [directional preposition] white truck 2. Black car 1 is [directional preposition] sign 1." 4. The second paragraph is a Proximal Template that describes spatial proximity relationships (e.g., near/far) between the same entities, using [proximal preposition] as placeholders. Example: "Black car 1 is [proximal preposition] white truck 2. Black car 1 is [proximal preposition] sign 1." 5. The numbers of placeholders in the above two paragraphs are the same, as they describe relationships for the same entities. 6. The number in each object name (e.g., black car 1, sign 2) indicates its horizontal left-to-right order in the image among entities of the same type. For vehicles, the order is based on all boxed vehicles in the second image. For signs and traffic lights, the order is based on their horizontal position in the image (from either image).
[Task and Requirements] 1. Based on the provided images and templates, predict a label INDEX (NOT the label string) from the following label lists for each [directional preposition] and [proximal preposition] placeholder. DIRECTIONAL_LABELS = ["at the back of", "at the back left of", "at the left of", "at the front left of", "at the front of", "at the front right of", "at the right of", "at the back right of"] PROXIMAL_LABELS = ["adjacent to", "close to", "at a distance", "far from", "far away from"] 2. Please output two integer lists: 'answers_directional' and 'answers_proximal' in a json format, where each entry is the index of the selected label from the provided label list. Output example: {{ "answers_directional": [4, 7, ..., 1], "answers_proximal": [2, 4, ..., 0] }} 3. The list answers_directional must contain only integer indices ranging from 0 to 7 (inclusive), and answers_proximal must contain indices from 0 to 4 (inclusive), as each index corresponds to a valid entry from the DIRECTIONAL_LABELS and PROXIMAL_LABELS lists, respectively. 4. The length of the answer lists must exactly match the number of [directional preposition] or [proximal preposition] placeholders in its respective template. This number will be explicitly provided to you along with the templates for each data example. 5. Do not provide explanations. Return only the two answer lists. 6. In some driving scenes, there may be no detectable entities, in which case both templates will contain only a single period ".". When this occurs, you should simply return two empty lists. The following are the two incomplete text paragraphs of this inference: [Directional Template] {template_directional} [Proximal Template] {template_proximal} The lists 'answers_directional' and 'answers_proximal' you return must each have EXACTLY {num_placeholders} entries. 'answers_directional' must contain ONLY INTEGERS FROM 0 to 7, and 'answers_proximal' must contain ONLY INTEGERS FROM 0 to 4. Do not use quotation marks around the numbers — all entries must be returned as raw INTEGERS, NOT STRINGS.

---

### C.3 Gaze Prediction

For gaze prediction, the general structure of this task is shown as Prompt. C.6. The detailed prompt we input is shown in Prompt. C.5. The original image and image with bbox of vehicles are shown in Fig. 8a and Fig. 8b, respectively. The output gaze attention map and gt attention map are shown in Fig. 10.

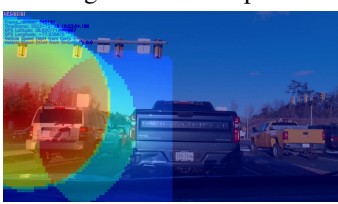 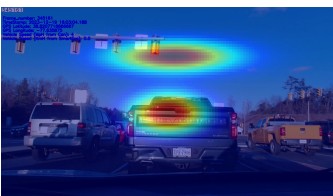 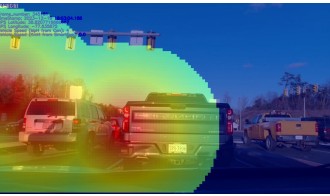

(a) Predicted human gaze attention map output by GPT-4o

(b) Predicted human gaze attention map output by Gemini-2.5-Pro

(c) GT human gaze attention map

Figure 10: Example of output attention maps for gaze prediction task

---

**Prompt C.5: Gaze Prediction Prompt Example**

You are an agent designed to predict the attention distribution of a human driver in the current driving scene, based on a real-time image and the gaze gray-scale from the previous 5 frames. Your prediction should reflect the driver's visual attention in the current frame.

[Task Summary] Please predict the human driver's attention on the current driving scene frame, based on the scene image and the driver's attention on the previous five frames.

[Input] 1. The first 5 inputs are grayscale heatmaps of the driver's attention on the previous 5 frames. Each has size (1080, 1920) and is downsampled by 16*16 average pooling, so each 16*16 patch contains the same value. 2. The 6th input is the full-resolution (1080, 1920) RGB image of the current driving scene. 3. Please consider the attention map from previous frames (the first five input images), and the content and their potential moving patterns in the current frame (the sixth input image), to make the best prediction on the driver's attention on the current frame.

[Output] 1. The output array should have shape (1080, 1920), reflecting the driver's visual attention on the current driving scene frame. 2. Please note that all values in the output array will be normalized into the range from 0 to 1, via Min-Max Normalization. 3. Instead of outputting the array directly, please generate a Python function that creates it using the code template below. Please output exactly in this format, ONLY modify the section between '# Modify here' and ' # End of modification'. 4. Code template: "'python import numpy as np  import cv2  def create_attn_map(previous_attn_list, current_img): attn_map = np.zeros(current_img.shape[:2], dtype=object) # Initialize the attention map # Modify here attn_map = todo # End of modification return cv2.normalize(attn_map, None, alpha=0, beta=1, norm_type=cv2.NORM_MINMAX, dtype=cv2.CV_32F) # Normalize the attention map via Min-Max Normalization "' 5. In the code template, 'previous_attn_maps' is a np.array (with size(5, 1080, 1920)) of five attention maps corresponding to the previous five frames, and 'current_img' is a np.array (with size (1080, 1920, 3)) of current driving scene image in BGR channel order.

[WARNING] 1. Output the entire Python function INDISE A MARKDOWN CODE BLOCK, starting with "'python and ending with "', as illustrated in the code template. Otherwise, the system will fail to extract your code and your output will be considered invalid. 2. DO NOT treat lists as NumPy arrays, they DON'T have attributes like '.astype', '.shape', '.dtype', or '.size'. Convert them into np.array if you need to apply these attribute. If violated, the output will be considered invalid. 3. When using OpenCV functions such as cv2.add, cv2.addWeighted, cv2.subtract, etc., MAKE SURE the input arrays have the same data type (e.g., both float32 or both uint8). If necessary, use .astype(np.float32) to convert them before the operation. Otherwise, it will cause cv2.errors.

[Important Requirements] 1. The output function can only rely on the two arguments given in the code template. 2. Please try to use only methods from the NumPy and OpenCV libraries. 3. Please do not define any additional helper functions outside 'create_attn_map'. 4. Please ensure that the code is syntactically correct, bug-free, and ready to run without modifications; try to keep your function concise and refined. 5. Please output only the Python function, without any additional explanatory text. 6. Please do NOT use 'cv2.saliency' module, it may not be available. 7. Please do NOT assume any existence of external files, functions, or methods.

**Prompt C.6: Gaze Prediction Example General**

Q: [Gaze Prediction Task Prompt] + [Python Code Template] + [Attention Map from Five Previous Frames] + [Original Image].

A:
```
import numpy as np import cv2 def create_attn_map(previous_attn_list,
current_img):  attn_map = np.zeros(current_img.shape[:2],
dtype=np.float32) # Initialize the attention map # Modify
here, assign value for 'attn_map' # Average the previous
attention maps avg_previous_attn = np.mean(previous_attn_list,
axis=0) # Focus on the road ahead and vehicles height, width
= current_img.shape[:2] road_region = current_img[height//2:,
:]  road_gray = cv2.cvtColor(road_region, cv2.COLOR_BGR2GRAY) _,
road_mask = cv2.threshold(road_gray, 127, 1, cv2.THRESH_BINARY)
# Combine the averaged previous attention with the road mask
attn_map[height//2:, :]  = avg_previous_attn[height//2:, :]  *
road_mask # End of modification return cv2.normalize(attn_map, None,
alpha=0, beta=1, norm_type=cv2.NORM_MINMAX, dtype=cv2.CV_32F) #
Normalize the attention map via Min-Max Normalization.
```

After a close look at the results in gaze prediction output by different models, we found that the widely used metrics such as person's correlation coefficient, KL-Divergence and information gain (IG) might not be able to fully reveal the performance of different MLLMs for gaze prediction tasks in our settings (given attention map from five previous frames). The main insight for giving human gaze attention map from five previous frames is we want the model to predict the human gaze attention map for current frame. In addition, since human gaze follows high spatial-temporal property, the ground-truth (GT) human gaze attention map should be the accumulation of these human gaze attention maps from 5 previous frames and current frame. Based on the results shown in Tab. 4, Qwen-VL-2.5-7B and InternVL-2.5-26B achieves the best and second best performance in human gaze prediction task. However, the output code of Qwen-VL2.5-7B and InternVL2.5-26B are always taking the average of human gaze attention map from five previous frames and do no additional operations on it, shown as Prompt. C.7. We don't think current metrics for gaze prediction can disclose the true capability of MLLMs while based on other model's output such as using operations such as gaussian blue or edge detection is also far away from the true gaze prediction. Thus, we claim that the MLLMs we test in our experiment shows little capability in predicting human gaze attention.

**Prompt C.7: Qwen-VL2.5-7B Output for Gaze Prediction Example**

```
import numpy as np import cv2 def create_attn_map(previous_attn_list,
current_img):  attn_map = np.zeros(current_img.shape[:2],
dtype=np.float32) # Initialize the attention map # Modify
here for i in range(len(previous_attn_list)):  attn_map +=
previous_attn_list[i] attn_map /= len(previous_attn_list) # End
of modification return cv2.normalize(attn_map, None, alpha=0,
beta=1, norm_type=cv2.NORM_MINMAX, dtype=cv2.CV_32F) # Normalize the
attention map via Min-Max Normalization
```

## C.4 Human-like Spatial Intelligence Grid Creation

---

**Prompt C.8: Human-like SIGC Example General**

Q: [Human-like SIGC task prompt] + [Original Image] + [Original Image with bbox of vehicles].
A:        {"vehicles":{"white car 1":[2,3],"gray truck 2":[5,2],"yellow truck 3":[7,3]},"traffic_lanes":{"lane 1":[[2,0],[2,9]],"lane 2":[[5,0],[5,9]],"lane 3":[[7,0],[7,9]]},"traffic_signs":{"sign 1":[3,1],"sign 2":[6,1]},"traffic_lights":{"light 1":[4,1],"light 2":[8,1]},"self":[4,0]}.

---

For human-like SIGC, it is similar to the task SIGC (Sec. C.1). The general structure of this task is shown as Prompt. C.8. The detailed prompt we input is shown in Prompt. C.9. The original image and image with bbox of vehicles are shown in Fig. 8a and Fig. 8b, respectively. The image size attention map is shown in Fig. 11a and SIG size attention map transformed using homographic transformation is shown in Fig. 11b. The attention weight of each object is the corresponding attention weight at their position in SIG. During evaluation, these attention weight will be used to scale the punishment of the incorrect prediction for different objects.

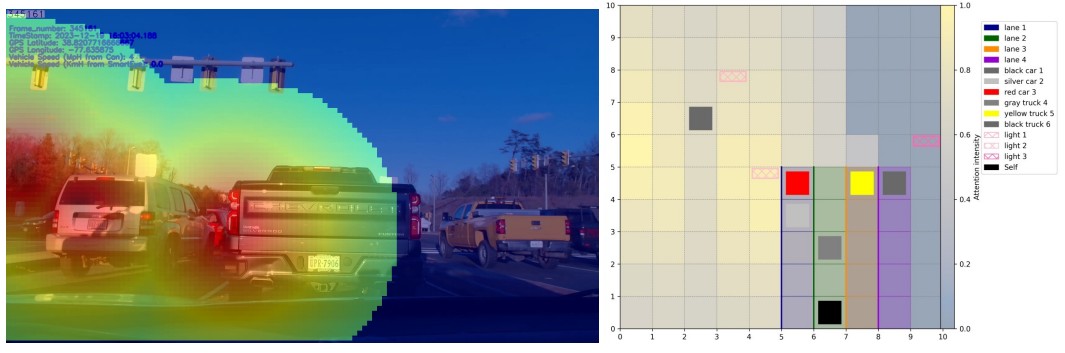

(a) Image size human gaze attention map      (b) SIG size human gaze attention map

Figure 11: Example of image and SIG size human gaze attention map

**Prompt C.9: Human-like SIGC Prompt Example**

[Task Summary] The first image captures an outdoor driving scene, and the second image is the same scene with bounding boxes on certain vehicles. Please identify vehicles, traffic lanes, traffic signs, and traffic lights within the image like a human, and understand the spatial arrangement of these entities. Specifically, assuming the bird's eye view of the scene is represented by a 10*10 grid, please estimate the center position of these entities within this grid. The output is expected to be a JSON file containing a dictionary.

[Task Details] <Overall> 1. The first image captures an outdoor driving scene, and the second image is the same scene but only with bounding boxes on certain vehicles. 2. Estimate the center positions of each instance within the first image, assuming the entire scene is represented by a 10*10 bird's eye view grid. 3. The entities to be estimated include: vehicles, traffic lanes, traffic signs, traffic lights, and the vehicle that capture the images (self). 4. Both the horizontal and vertical coordinates of the grid range from 0 to 9, so all estimated positions (e.g., [x_1, y_1]) must fall within this range. 5. Estimated location of each instance should accurately reflect its real position in the scene, preserving the relative spatial relationships among all objects. 6. Please be aware of the front-backward or left-right relationship between instances, as there will be partial occlusion. 7. Since it is a bird's eye view grid, for all instances, more far away objects should be placed in the higher row number and the closer objects should be placed in the lower row number. <Vehicles> 1. Detect vehicles in the first image that are enclosed by bounding boxes in the second image only, and estimate the center positions of these enclosed vehicles within the grid. 2. The output is a key-value pair, which is expected to be exactly like: "vehicles": {"black car 1": [x_1, y_1], "gray truck 2": [x_2, y_2], "yellow bus 3": [x_3, y_3], "blue van 4": [x_4, y_4], ...}, where each vehicle instance is named by color, vehicle type, and order. 3. The color of vehicles are summarized into: gray, black, white, silver, blue, green, yellow, red, purple. Other colors need to be attributed to the one closest to these colors. 4. The type of vehicles must be classified into exactly four categories: "car" (which include sedans, hatchbacks, and suvs), "truck" (including pickup trucks), "van", and "bus". Do not use "suv" or any other subcategory names directly as a type. Instead, map them into one of the four allowed categories. 5. The order of vehicles must be assigned globally, based strictly on their horizontal position in the first image. The left-most vehicle in the image must be numbered "1", the second left-most vehicle must be numbered "2", and so on. This numbering should apply across all vehicles, without restarting for different colors or types. 6. Vehicles usually travel within traffic lanes, so the estimated position of a vehicle moving in the same direction as the "self" vehicle should typically fall within the coordinates of a corresponding lane. This does not apply to vehicles traveling in the other directions or that are stationary. <Traffic lanes> 1. Detect all traffic lanes in the same driving direction as the vehicle capturing the image, and estimate the lane position within the grid. One lane can be represented by multiple adjacent points as it is long. 2. The output is a key-value pair, which is expected to be exactly like: "traffic_lanes": {"lane 1": [[x_11, y_11], [x_12, y_12], ...], "lane 2": [[x_21, y_21], [x_22, y_22], ...], ...}, where each lane is named by order. 3. The order of traffic lanes is based strictly on their horizontal position in the first image. The left-most lane in the image must be numbered "1", the second left-most lane must be numbered "2", and so on. 4. Each lane is typically straight, so the horizontal coordinates of its points should usually be the same, forming a vertical line in the grid, unless the lane is clearly turning, merging, or splitting. 5. Additionally, adjacent lanes are typically close together, so their horizontal coordinate values of adjacent lanes should usually differ by only 1. <Traffic signs> 1. Detect all traffic signs in the first image, and estimate the center positions of these traffic signs. 2. The output is a key-value pair, which is expected to be exactly like: "traffic_signs": {"sign 1": [x_1, y_1], "sign 2": [x_2, y_2], ...}, where each sign is named by order. 3. The order of traffic signs is based strictly on their horizontal position in the first image. The left-most sign in the image must be numbered "1", the second left-most sign must be numbered "2", and so on. 4. If multiple signs are mounted on the same horizontal pole, please treat them as separate instances. <Traffic lights> 1. Detect all traffic lights in the first image, and estimate the center positions of these traffic lights. 2. The output is a key-value pair, which is expected to be exactly like: "traffic_lights": {"light 1": [x_1, y_1], "light 2": [x_2, y_2], ...}, where each light is named by order. 3. The order of traffic lights is based strictly on their horizontal position in the first image. The left-most light in the image must be numbered "1", the second left-most light must be numbered "2", and so on. 4. If multiple traffic lights are mounted on the same horizontal pole, please treat them as one single light and use the midpoint between them as the center. <Self> 1. Estimate the center location of the vehicle that captures the image. 2. The output is expected to be exactly like: '"self": [x, 0]' 3. The vehicle capturing the image should be placed in the point with raw index 0 and with column index depends on different images.

[Output] 1. Combine the key-value pairs of vehicles, traffic lanes, traffic signs and traffic lights into one dictionary, as final output. 2. The final output dictionary is expected to be exactly like: { "vehicles": {"black car 1": [x_1, y_1], "gray truck 2": [x_2, y_2], "yellow bus 3": [x_3, y_3], "blue van 4": [x_4, y_4], ...}, "traffic_lanes": {"lane 1": [[x_11, y_11], [x_12, y_12], ...], "lane 2": [[x_21, y_22], ...], ...}, "traffic_signs": {"sign 1": [x_1, y_1], "sign 2": [x_2, y_2], ...}, "traffic_lights": {"light 1": [x_1, y_1], "light 2": [x_2, y_2], ...}, "self": [x, 0] } 3. Please output the final dictionary in pure JSON format, without any additional text or explanation before or after. 4. All keys and string values in the output dictionary must be in lowercase letters only.

### C.5 Human-like Spatial Relation Paragraph Filling

---

**Prompt C.10: Human-like SRPF Example General**

Q: [SRPF task prompt] + [Original Image] + [Original Image with bbox of vehicles] + [Directional Relation Paragraph Template] + [Proximal Relation Paragraph Template].
A: {"answers_directional":[2,2,4,4,4,4,4,4,2,4,4,4,4,4,4,4,4,4,4,4,4,4,6,4,4,4,4,4,4,4,4,4,4,4,4],
"answers_proximal":[0,0,1,1,1,2,2,2,0,1,1,1,2,2,2,1,1,1,2,2,2,1,1,2,2,2,1,1,2,2,2,2,2,2,2]}.

---

For human-like SRPF, it is similar to the task SRPF (Sec. C.2). The general structure of this task is shown as Prompt. C.10. The detailed prompt we input is shown in Prompt. C.11. The original image and image with bbox of vehicles are shown in Fig. 8a and Fig. 8b, respectively. The image size attention map is shown in Fig. 11a and SIG size attention map transformed using homographic transformation is shown in Fig. 11b. The attention weight for each sentence between two objects would be the average of these two objects using their corresponding attention weight at their position in SIG.

---

**Prompt C.11: Human-like SRPF Prompt Example**

[Input Description] You will be provided with two images and two incomplete text paragraphs, think as a human driver. 1. The first image shows a real-world driving scene. 2. The second image is the same scene with bounding boxes drawn on several vehicles. 3. The first paragraph is a Directional Template that describes the directional relationships between certain vehicles, traffic signs, or traffic lights, using [directional preposition] as placeholders. Example: "Black car 1 is [directional preposition] white truck 2. Black car 1 is [directional preposition] sign 1." 4. The second paragraph is a Proximal Template that describes spatial proximity relationships (e.g., near/far) between the same entities, using [proximal preposition] as placeholders. Example: "Black car 1 is [proximal preposition] white truck 2. Black car 1 is [proximal preposition] sign 1." 5. The numbers of placeholders in the above two paragraphs are the same, as they describe relationships for the same entities. 6. The number in each object name (e.g., black car 1, sign 2) indicates its horizontal left-to-right order in the image among entities of the same type. For vehicles, the order is based on all boxed vehicles in the second image. For signs and traffic lights, the order is based on their horizontal position in the image (from either image).
[Task and Requirements] 1. Based on the provided images and templates, predict a label INDEX (NOT the label string) from the following label lists for each [directional preposition] and [proximal preposition] placeholder. DIRECTIONAL_LABELS = ["at the back of", "at the back left of", "at the left of", "at the front left of", "at the front of", "at the front right of", "at the right of", "at the back right of"] PROXIMAL_LABELS = ["adjacent to", "close to", "at a distance", "far from", "far away from"] 2. Please output two integer lists: 'answers_directional' and 'answers_proximal' in a json format, where each entry is the index of the selected label from the provided label list. Output example: {{ änswers_directional": [4, 7, ..., 1], änswers_proximal": [2, 4, ..., 0] }} 3. The list answers_directional must contain only integer indices ranging from 0 to 7 (inclusive), and answers_proximal must contain indices from 0 to 4 (inclusive), as each index corresponds to a valid entry from the DIRECTIONAL_LABELS and PROXIMAL_LABELS lists, respectively. 4. The length of the answer lists must exactly match the number of [directional preposition] or [proximal preposition] placeholders in its respective template. This number will be explicitly provided to you along with the templates for each data example. 5. Do not provide explanations. Return only the two answer lists. 6. In some driving scenes, there may be no detectable entities, in which case both templates will contain only a single period ".". When this occurs, you should simply return two empty lists. The following are the two incomplete text paragraphs of this inference: [Directional Template] {template_directional} [Proximal Template] {template_proximal} The lists 'answers_directional' and 'answers_proximal' you return must each have EXACTLY {num_placeholders} entries. 'answers_directional' must contain ONLY INTEGERS FROM 0 to 7, and 'answers_proximal' must contain ONLY INTEGERS FROM 0 to 4. Do not use quotation marks around the numbers — all entries must be returned as raw INTEGERS, NOT STRINGS.

---

