# OpenReview forum: "Towards Physics-informed Spatial Intelligence with Human Priors: An Autonomous Driving Pilot Study"
_NeurIPS.cc/2025/Conference — NeurIPS 2025 spotlight_

### Official Review · Reviewer_BBAE · 2025-06-28

**Clarity:** 2
**Significance:** 3
**Originality:** 3
**Rating:** 5
**Confidence:** 2

**Summary:**

The authors introduce Spatial Intelligence Grid (SIG), a new format for representing autonomous driving scenes, where the objects as well as their geometrical spatial relationships are embedded into a grid-based data schema. They create a dataset of input scenes and annotated SIGs and define the tasks of extracting its essential components, which include the birds-eye grid view of the scene featuring the objects and their locations, and the spatial relation graph (SRG) and the spatial relation paragraph (SRP), which describe the spatial relations between objects in a numeric and linguistic terms, respectively. They also define the metrics used to evaluate the model performance on these tasks, which the authors claim to more accurately reflect the VSI capabilities of models. Finally, the authors show that these representations are more effective for few-shot in context learning than the standard VQA style representations.

**Questions:**

- How are gamma and beta for weighted similarity chosen?
- Why wasn’t gaze based weighting also applied onto the MLSM?
- Why does the fact that the relative ranking of models doesn’t change significantly with the addition of gaze based weighting point to the conclusion that the “current MLLMs are still unable to interpret the prioritization of different objects in a scene”?
- Any insights on why Gemini-2.5-Pro is so much better than GPT-4o on ICL on SIGBench-tiny, despite not being that much better on SIGBench itself? It might have been interesting to see the performance of other models on ICL (e.g. if it turned out an open source model was competitive), given that it appears to be not all that tied to the SIGBench performance

**Ethical Concerns:**

["NO or VERY MINOR ethics concerns only"]

**Final Justification:**

I appreciate the replies provided by the authors. I will keep the ranking from the initial review.

**Limitations:**

yes

**Quality:**

3

**Strengths And Weaknesses:**

Strengths:
- The paper introduces a novel dataset, featuring a new way of representing autonomous driving scenes, which can be valuable to the research community
- The authors evaluate a range of state-of-the art VLMs on their tasks, simultaneously demonstrating both the utility of the resulting representations by showing them to be more effective for 3-shot in context learning than standard VQA based approaches, and showing a considerable existing performance gap to humans, motivating further research
- Rich appendix with detailed descriptions of their data collection and annotation process, as well as examples of their proposed tasks

Weaknesses:
- The authors claim to “derive a set of SIG-based optimal evaluation metrics that rigorously quantify a model’s true VSI capabilities”, and while it does make sense that these metrics point to the VSI capabilities of models, their "optimality" feels overstated. In general, I do not think that the work clearly demonstrates the advantage of these evaluation metrics in terms of representing the spatial intelligence of models
- The choice of the 3 context samples could have had some degree of impact. It would have been interesting to see if this was negligible or not
- I feel like the results on human-like VSI offer too little insights compared to how much space in the main paper was dedicated to it. On a more minor note, in some cases, less important information, such as Eq 7 in 4.1 (which is also featured in the appendix) is presented in the main paper, while on the other hand, more important details are omitted (e.g. in 3.3 where it leaves out how the correspondences between image pixels and grid cells were established when calculating the homographic matrix)

---

> ### Author Rebuttal · Authors · 2025-07-29
>
> We want to thank the reviewer for your efforts in handling this submission and we appreciate you acknowledging novelty and utility of our proposed SIG and the rich appendix with detailed description. We will address your questions one by one in following.
>
> **W-1 Optimality is overstated.** We agree that “optimality” is stronger than what our current evidence supports. We will refine the wording to: "derive a set of \textbf{SIG-informed} evaluation...". This phrasing better reflects how our metrics leverage the SIG to assess spatial reasoning without overclaiming optimality.
>
> **W-2 Choice of 3 context samples have impacts.** To answer this question, we conduct extensive experiments investigating the effect brought by the choice of training samples for ICL on GPT-4o. The results are shown in the following table. As you can see, by more carefully selecting the training samples, the performance of ICL is much higher compared to random data sample selection we mentioned in our paper. These results further demonstrate the potential of SIG as a new data schema for improving visual-spatial intelligence of MLLM.
>
> *Table. Quantitative comparison of 3-shot ICL for grid-based VSI tasks on SIGBench-tiny with data selection based on number of objects on GPT-4o. Partial-empty means the number of objects in one or two of [vehicles, traffic signs, traffic lights] equals 0, while non-empty means the number of objects in vehicles, traffic signs and traffic lights in each data sample $\geq 1$. Results in Random is copied from the line GPT-4o ICL-SIG in Tab. 3 of our submission.*
> | Type               | #Objects | **MLSM** |     |     |        | **SRGS** |     | **SRD (Directional)** |     |        | **SRD (Proximal)**   |     |        |
> |--------------------|:--------:|:--------:|:---:|:---:|:------:|:--------:|:---:|:---------------------:|:---:|:------:|:--------------------:|:---:|:------:|
> |                    |          | P↑       | R↑  | F1↑ | AssA↑  | S↑       | WS↑ | MAE↓                 | MSE↓| Acc↑  | MAE↓                | MSE↓| Acc↑  |
> | **Random**         | Random   | 0.592    | 0.438| 0.479 | 0.328 | 0.337    | 0.357| 1.593                | 4.094| 0.220 | 0.775               | 1.271| 0.436 |
> | **Partial‑empty**  | 3        | 0.582    | 0.398| 0.459 | 0.310 | 0.303    | 0.332| 1.981                | 5.563| 0.165 | 0.775               | 1.231| 0.419 |
> |                    | 4        | 0.562    | 0.405| 0.458 | 0.314 | 0.326    | 0.341| 1.777                | 4.667| 0.180 | 0.845               | 1.314| 0.367 |
> |                    | 5        | 0.552    | 0.393| 0.446 | 0.306 | 0.306    | 0.324| 1.839                | 5.004| 0.191 | 0.832               | 1.288| 0.375 |
> |                    | 8        | 0.571    | 0.465| 0.499 | 0.349 | 0.352    | 0.365| 1.668                | 4.340| 0.203 | 0.797               | 1.244| 0.393 |
> |                    | 10       | 0.563    | 0.457| 0.487 | 0.337 | 0.326    | 0.334| 1.671                | 4.471| 0.229 | 0.757               | 1.159| 0.429 |
> | **Non‑empty**      | 3        | 0.561    | 0.382| 0.443 | 0.300 | 0.316    | 0.331| 1.980                | 5.560| 0.155 | 0.876               | 1.442| 0.368 |
> |                    | 4        | 0.584    | 0.419| 0.476 | 0.331 | 0.340    | 0.357| 1.881                | 5.164| 0.167 | 0.760               | 1.125| 0.398 |
> |                    | 5        | 0.572    | 0.417| 0.469 | 0.321 | 0.349    | 0.361| 1.921                | 5.362| 0.184 | 0.783               | 1.236| 0.412 |
> |                    | 8        | 0.543    | 0.484| 0.499 | 0.350 | 0.362    | 0.365| 1.667                | 4.359| 0.204 | 0.848               | 1.313| 0.357 |
> |                    | 10       | 0.533    | 0.460| 0.490 | 0.339 | 0.357    | 0.362| 1.690                | 4.537| 0.212 | 0.850               | 1.467| 0.403 |
> | **Combined**       | 3+4+5    | 0.567    | 0.411| 0.464 | 0.317 | 0.346    | 0.354| 1.885                | 5.143| 0.150 | 0.824               | 1.304| 0.372 |
> |                    | 3+4+8    | 0.600    | 0.442| 0.501 | 0.350 | 0.365    | 0.380| 1.849                | 5.000| 0.171 | 0.780               | 1.241| 0.421 |
> |                    | 4+5+8    | 0.571    | 0.426| 0.477 | 0.330 | 0.363    | 0.373| 1.665                | 4.335| 0.204 | 0.732               | 1.046| 0.419 |
> |                    | 4+8+10   | 0.598    | 0.455| 0.505 | 0.353 | 0.360    | 0.374| 1.566                | 4.007| 0.229 | 0.779               | 1.253| 0.430 |
>
> **W-3 Human-like VSI offers little insights.** We apologize for omitting certain implementation details due to space constraints. \
> Our primary reason for including human‑like VSI is to capture selective attention in driving: human drivers naturally focus on a small subset of scene elements—such as a red traffic light or a pedestrian stepping off the curb, when making safety‑critical decisions, rather than processing every object equally. Incorporating gaze‑derived saliency into SIG allows us to weight model errors according to human importance, offering a more systematic evaluation of decision‑relevant understanding, a feature currently absent in existing AD benchmarks. Our attention-based metrics (SRGS, SRD) not only assess whether a model can reconstruct spatial layouts, but also whether it can prioritize the right elements, mirroring the attention mechnism of a safe driver. This human‑centered weighting transforms SIGBench from a general VSI benchmark into one that evaluates the  model’s spatial understanding in real‑world driving contexts. \
> We will provide detailed homographic transformation in Appendix.
>
> **Q-1 Values of gamma and beta.** We set $$ \gamma = \frac{2}{|\mathcal{V}|}, \quad \beta = \frac{1}{|\mathcal{E}|} $$
> where $|\mathcal{V}|$ and $|\mathcal{E}|$ are the numbers of nodes and edges in the ground-truch SIG, respectively. This 1) bounds and scales insertion/deletion costs across graphs, and 2) weights edge operations more heavily (2:1) than node operations to reflect their inherently greater impact on GED.
>
> **Q-2 Why not attention-based MLSM?** In designing our human‑like VSI metrics, we choose to apply gaze weighting to SRGS rather than MLSM for two key reasons:
> - Metric focus: 1) MLSM evaluates the absolute placement of each object on the grid, i.e. how closely a predicted cell coordinate matches its GT; 2) SRGS assesses the relative structure of the scene graph. Human gaze naturally highlights relational importance ("Which objects matter most in context?") rather than precise metric distances. By weighting SRGS, we ensure that errors among highly attended object relationships incur larger penalties, while preserving MLSM’s role as an unbiased localization check.
> - Artistic analogy: Consider an artist sketching a scene: they first lay out all object contours (the "big picture") before adding relational details where attention is focused. In this process, they focus on all objects inside the attention range instead of pixel-level or object-level information. Thus, we think attention-based SRGS is more aligned to the evaluation of human-like VSI.
>
> **Q-3 Why does relative ranking of MLLMs doesn't change significantly point to MLLMs are still unable to prioritize objects?** Intuitively, if a model truly understood which objects garner human attention, penalizing errors on those high‑saliency objects more heavily would cause its performance to improve relative to others, and thus alter the ranking of models under gaze‑weighted metrics. In other words, a model that prioritizes the "right" objects would gain a bigger boost when those objects are weighted more. Because we observe that the order of models remains essentially unchanged after applying gaze‑based weights, this implies that none of the evaluated MLLMs consistently emphasize high‑attention elements over low‑attention ones. In turn, this stability in ranking indicates that current MLLMs have yet to internalize human‑like prioritization of scene objects.
>
> **Q-4 Insights on results of ICL on SIGBench-tiny and ICL using open-source models.** The zero‑shot rankings on SIGBench and SIGBench‑Tiny are consistent: Gemini‑2.5‑Pro leads on MLSM and GPT‑4o on SRGS. Their ICL performance gains diverges significantly on SIGBench‑Tiny split. Our insight for this occurrence is that if a MLLM can determine the absolute spatial position of object correctly (revealed from metrics in MLSM), it's easy for it to improve the understanding of relative spatial relationship between objects. However, to improve the localization of objects in 3D space requires further training and fine-tuning. For open-source models, we conduct the ICL using Qwen-VL-7B and Qwen-VL-32B on SIGBench-Tiny. The following results demonstrates both models are acquire significantly improvements on MLSM and SRGS metrics, highlighting SIG's strong potential for improving VSI.  Our SIG representation can also be generalized to other domains besides AD, please refer to reviewer **wMQv W-1&Q-2**.
>
> | Models             | Type    | **MLSM** |   |   |     | **SRGS** |   | **SRD (Directional)** |   |   | **SRD (Proximal)**   |   |   |
> |--------------------|---------|:--------:|:-:|:-:|:----:|:--------:|:-:|:---------------------:|:-:|:-:|:--------------------:|:-:|:-:|
> |                    |         | P↑       | R↑ | F1↑| AssA↑| S↑       | WS↑| MAE↓                  | MSE↓| Acc↑| MAE↓                 | MSE↓| Acc↑|
> | Qwen‑VL 2.5‑7B| Z‑S     | 0.273   | 0.331 | 0.283 | 0.173 | 0.104   | 0.080 | 2.129         | 6.084  | 0.093  | 1.230           | 2.461  | 0.247  |
> |                    | ICL‑SIG | 0.470   | 0.340 | 0.371 | 0.241 | 0.216   | 0.217 | 1.691         | 4.372  | 0.204  | 1.271           | 2.525  | 0.216  |
> | Qwen‑VL 2.5‑32B| Z‑S     | 0.455    |0.369|0.397|0.265|0.232     |0.207|1.767                |4.442|0.144|1.718                 |4.000|0.115|
> |                    | ICL‑SIG | 0.569    |0.468|0.499|0.351|0.328     |0.347|2.161                 |6.120|0.100|1.447                 |3.183|0.199|

---

> > ### Comment · Reviewer_BBAE · 2025-08-04
> >
> > Thank you for the reply. Regarding the results of 3-shot ICL experiments, I appreciate the detailed analysis on the choice of the ICL examples. Thile I'd be inclined to believe that ICL-SIG would on average perform better than ICL-MC and zero-shot across different choices of examples, it is also apparent that the results can vary significantly with different choices of examples (for instance, certain entries such as partially empty 3, 4 and 5 and Non-empty 3 from your reply achieve a lower F1 than zero shot). While it does not really cast doubt on the utility of SIG in this context, the results in Table 3 would benefit from error bars computed over multiple re-runs with different random choices of examples.
> > Regarding the conclusion about the models' ability to prioritize salient objects that was drawn from the lack of change in relative ranking from gaze based weighting - while I think this explanation may be plausible, I don't think it is really proven.
> > I appreciate the answers that the authors gave to the remaining questions, I don't have any additional comments on them.

---

> ### Author Response · Authors · 2025-08-03
> **Update the complete results of SIG-COCO and SIG-ARKitScenes and results of new data sample selection strategy for SIG-based ICL**
>
> **W-2 Choice of 3 context samples have impacts.** We further investigate ICL with adaptive SIG samples. Different from heterogeneous SIG samples, which present each shot with a distinct image and its GT SIG, adaptive SIG samples uses the same image across multiple shots, pairing each predicted SIG with the GT SIG for direct comparison. Specifically, we select and rank several candidate SIGs for each image by their precision scores under the MLSM metric, then present these ranked SIGs as few‐shot examples. Intuitively, exposing the model to both high‐ and low‐quality SIG should sharpen its ability to discern and leverage the best spatial descriptions of the image and convey it using SIG. This strategy mirrors human learning: comparing several imperfect attempts against a correct one sharpens the ability to discern and internalize the optimal solution. The results are shown in the following Table. For MLSM and SRGS, ICL using adaptive SIG samples delivers substantial gains over using heterogeneous SIG samples from 3- to 6-shot ICL. This means when the number of data samples is relatively less, adaptive SIG samples can bring better improvements in model's VSI. However, for 9-shot ICL, heterogeneous SIG samples continue to yield further improvements, whereas adaptive SIG samples plateau over 6-shot, which indicates adaptive SIG samples' limited scalability compared to heterogeneous SIG samples. These occurrences underscore the principle that, while in-depth exploration of a single SIG sample is valuable, achieving optimal VSI also requires learning from a broad diversity of instances. For SRD, adaptive SIG samples consistently achieve better performance than using heterogeneous SIG samples. Taken together, these findings indicate that neither strategy alone is sufficient. To maximize the performance of SIG‐based ICL, we should combine adaptive and heterogeneous SIG samples.
>
> *Table. Quantitative comparison of few shot ICL for grid-based VSI tasks on SIGBench-tiny with data selection based on adaptive SIG on GPT-4o. Different means we use different image and its corresponding GT SIG (e.g. 3-shot uses 3 different images), while adaptive means using the same image with different predicted SIG and GT SIG (e.g. 3-shot uses the same image with 3 different predicted SIG).*
> | Shots | Type      | **MLSM** |        |        |        | **SRGS** |       | **SRD (Directional)** |        |        | **SRD (Proximal)** |        |        |
> |-------|-----------|:--------:|:------:|:------:|:------:|:--------:|:-----:|:---------------------:|:------:|:------:|:------------------:|:------:|:------:|
> |       |           | P↑       | R↑     | F1↑    | AssA↑  | S↑       | WS↑   | MAE↓                  | MSE↓   | Acc↑   | MAE↓               | MSE↓   | Acc↑   |
> | **3** | Different | 0.561    | 0.415  | 0.460  | 0.318  | 0.329    | 0.343 | 1.673                | 4.379  | 0.211  | 0.722              | 1.044  | 0.422  |
> |       | Adaptive  | 0.557    | 0.391  | 0.445  | 0.300  | 0.323    | 0.336 | 1.559                | 3.867  | 0.220  | 0.694              | 1.008  | 0.444  |
> | **6** | Different | 0.570    | 0.417  | 0.469  | 0.320  | 0.331    | 0.346 | 1.563                | 3.933  | 0.223  | 0.699              | 0.995  | 0.429  |
> |       | Adaptive  | 0.572    | 0.418  | 0.470  | 0.323  | 0.339    | 0.347 | 1.485                | 3.637  | 0.232  | 0.671              | 0.949  | 0.453  |
> | **9** | Different | 0.588    | 0.447  | 0.499  | 0.346  | 0.364    | 0.379 | 1.607                | 4.040  | 0.213  | 0.657              | 0.954  | 0.470  |
> |       | Adaptive  | 0.573    | 0.420  | 0.471  | 0.325  | 0.341    | 0.350 | 1.598                | 4.018  | 0.204  | 0.659              | 1.019  | 0.484  |
>
> Please refer to our new comment to reviewer **wMQv W-1&Q-2** for the udpated results on SIG-COCO and SIG-ARKitScenes, which demonstrates the strong generalizability of our proposed SIG to domains other than autonomous driving scenario.

---

> ### Author Response · Authors · 2025-08-05
> **Error bar statistics of multi-run ICL using GPT-4o**
>
> **Error Bar Statistics of Multi-run ICL.** Thank you for your expertise—your review and comments has further strengthened our work. We re-run ICL with random data sample selection on SIGBench-tiny for 5 times using GPT-4o and Gemini-2.5-Pro. The error bar statistics are listed in the following tables. Due to space limitation, we separate two tables to two comments.
>
>
> *Table. Error bar statistics of ICL on GPT-4o using random data selection with 5 run on SIGBench-tiny. ICL-MC meaning ICL using multiple-choice VQA and ICL-SIG meaning ICL using SIG. Std means standard deviation and Sem is standard error of mean.*
> | Type | Metrics | **MLSM** |        |        |        | **SRGS** |       | **SRD (Directional)** |        |        | **SRD (Proximal)** |        |        | MC (ACC)  |   |
> |-------|-----------|:--------:|:------:|:------:|:------:|:--------:|:-----:|:---------------------:|:------:|:------:|:------------------:|:------:|:------:|:------:|:------:|
> |       |           | P↑       | R↑     | F1↑    | AssA↑  | S↑       | WS↑   | MAE↓                  | MSE↓   | Acc↑   | MAE↓               | MSE↓   | Acc↑   | Dir.↑   | Prox.↑   |
> | **ICL-MC** | Mean | 0.522       | 0.418       | 0.465        | 0.318          | 0.322       | 0.324        | 1.667                      | 4.291                      | 0.220                      | 0.930                     | 1.558                     | 0.343                     | 0.139              | 0.329               |
> |       | Std  | 0.172       | 0.183       | 0.162        | 0.142          | 0.211       | 0.196        | 0.812                      | 3.339                      | 0.227                      | 0.575                     | 1.481                     | 0.298                     | 0.212              | 0.286               |
> |  | Sem | 0.018       | 0.019       | 0.017        | 0.015          | 0.022       | 0.021        | 0.086                      | 0.352                      | 0.024                      | 0.061                     | 0.156                     | 0.031                     | 0.022              | 0.030               |
> |       | Min  | 0.091       | 0.073       | 0.081        | 0.042          | 0.000       | 0.000        | 0.000                      | 0.000                      | 0.000                      | 0.000                     | 0.000                     | 0.000                     | 0.000              | 0.000               |
> |       | Max  | 1.000       | 0.909       | 0.864        | 0.760          | 0.955       | 0.897        | 4.000                      | 16.000                     | 1.000                      | 3.000                     | 9.000                     | 1.000                     | 1.000              | 1.000               |
> |       | Median  | 0.523       | 0.411       | 0.455        | 0.308          | 0.327       | 0.328        | 1.667                      | 3.912                      | 0.163                      | 0.920                     | 1.143                     | 0.333                     | 0.000              | 0.250               |
> | **ICL-SIG** | Mean | 0.583       | 0.438       | 0.471        | 0.325          | 0.335       | 0.349        | 1.586                      | 4.032                      | 0.221                      | 0.762                     | 1.194                     | 0.423                     | 0.157              | 0.303               |
> |       | Std  | 0.176       | 0.177       | 0.166        | 0.147          | 0.198       | 0.181        | 0.825                      | 3.170                      | 0.255                      | 0.551                     | 1.361                     | 0.313                     | 0.222              | 0.278               |
> |       | Sem | 0.019       | 0.019       | 0.018        | 0.015          | 0.021       | 0.019        | 0.087                      | 0.334                      | 0.027                      | 0.058                     | 0.143                     | 0.033                     | 0.024              | 0.030               |
> |       | Min  | 0.115       | 0.068       | 0.105        | 0.055          | 0.000       | 0.000        | 0.000                      | 0.000                      | 0.000                      | 0.000                     | 0.000                     | 0.000                     | 0.000              | 0.000               |
> |       | Max  | 1.000       | 1.000       | 0.909        | 0.833          | 0.987       | 0.967        | 4.000                      | 16.000                     | 1.000                      | 3.000                     | 9.000                     | 1.000                     | 1.000              | 1.000               |
> |       | Median  | 0.594       | 0.415       | 0.475        | 0.331          | 0.340       | 0.352        | 1.500                      | 3.673                      | 0.167                      | 0.667                     | 0.713                     | 0.360                     | 0.000              | 0.250               |

---

> ### Author Response · Authors · 2025-08-05
> **Error bar statistics of multi-run ICL using Gemini-2.5-Pro**
>
> *Table. Error bar statistics of ICL on Gemini-2.5-Pro using random data selection with 5 run.*
> | Type | Metrics | **MLSM** |        |        |        | **SRGS** |       | **SRD (Directional)** |        |        | **SRD (Proximal)** |        |        | MC (ACC)  |   |
> |-------|-----------|:--------:|:------:|:------:|:------:|:--------:|:-----:|:---------------------:|:------:|:------:|:------------------:|:------:|:------:|:------:|:------:|
> |       |           | P↑       | R↑     | F1↑    | AssA↑  | S↑       | WS↑   | MAE↓                  | MSE↓   | Acc↑   | MAE↓               | MSE↓   | Acc↑   | Dir.↑   | Prox.↑   |
> | **ICL-MC** | Mean | 0.559    | 0.633    | 0.581    | 0.422    | 0.287    | 0.296    | 1.331                 | 3.247                 | 0.293                 | 0.733               | 1.082               | 0.426               | 0.157       | 0.373       |
> |       | Std  | 0.155    | 0.151    | 0.132    | 0.133    | 0.267    | 0.248    | 0.897                 | 3.360                 | 0.264                 | 0.457               | 0.926               | 0.301               | 0.242       | 0.291       |
> |  | Sem | 0.016    | 0.016    | 0.014    | 0.014    | 0.028    | 0.026    | 0.095                 | 0.354                 | 0.028                 | 0.048               | 0.098               | 0.032               | 0.025       | 0.030       |
> |       | Min  | 0.104    | 0.133    | 0.132    | 0.070    | 0.000    | 0.000    | 0.000                 | 0.000                 | 0.000                 | 0.000               | 0.000               | 0.000               | 0.000       | 0.000       |
> |       | Max  | 0.939    | 1.000    | 0.939    | 0.886    | 0.902    | 0.904    | 4.000                 | 16.000                | 1.000                 | 2.000               | 4.300               | 1.000               | 1.000       | 1.000       |
> |       | Median  | 0.552    | 0.651    | 0.586    | 0.414    | 0.261    | 0.292    | 1.000                 | 1.750                 | 0.293                 | 0.707               | 0.821               | 0.347               | 0.000       | 0.333       |
> | **ICL-SIG** | Mean | 0.611    | 0.629    | 0.610    | 0.453    | 0.381    | 0.384    | 0.991                 | 2.128                 | 0.394                 | 0.652               | 0.887               | 0.455               | 0.299       | 0.420       |
> |       | Std  | 0.155    | 0.161    | 0.140    | 0.147    | 0.270    | 0.244    | 0.743                 | 2.596                 | 0.283                 | 0.423               | 0.790               | 0.300               | 0.273       | 0.296       |
> |       | Sem | 0.016    | 0.017    | 0.015    | 0.016    | 0.029    | 0.026    | 0.118                 | 0.385                 | 0.049                 | 0.073               | 0.130               | 0.053               | 0.029       | 0.031       |
> |       | Min  | 0.115    | 0.115    | 0.115    | 0.061    | 0.000    | 0.000    | 0.000                 | 0.000                 | 0.000                 | 0.000               | 0.000               | 0.000               | 0.000       | 0.000       |
> |       | Max  | 1.000    | 1.000    | 1.000    | 1.000    | 1.000    | 1.000    | 4.000                 | 16.00                | 1.000                 | 2.000               | 4.100               | 1.000               | 1.000       | 1.000       |
> |       | Median  | 0.612    | 0.650    | 0.616    | 0.445    | 0.395    | 0.386    | 1.000                 | 1.327                 | 0.333                 | 0.667               | 0.667               | 0.434               | 0.250       | 0.500       |

---

> > ### Comment · Reviewer_BBAE · 2025-08-06
> > **Re: Error bar statistics of multi-run ICL using GPT-4o/Gemini-2.5-Pro**
> >
> > I appreciate the additional re-runs. It does, indeed, seem like the choice of examples can have significant impact on the results. Just a quick question, how was Sem computed, because it appears quite low for 5 re-runs (just an example is ok).

---

> ### Author Response · Authors · 2025-08-05
> **Attention-based MLSM result**
>
> **Models' ability to prioritize salient objects.** Thanks for your helpful feedback. We agree that our current results cannot completely prove that MLLMs do not have the ability to prioritize the object in a scene. We also implement the attention-based MLSM (Human-like MLSM) and the results are shown in the following table. The ranking of MLLMs in MLSM and H-MLSM are still similar, we hope this results can bring some more insights.
>
> *Table. Quantitative comparison of different MLLMs on SIGBench for MLSM and Human-like MLSM (H-MLSM). We incorporate human attention into MLSM to create H-MLSM. The MLSM results are borrowed from Tab. 2 in our manuscript for comparison.*
> | Type | Metrics | **MLSM** |        |        |        | **H-MLSM** |        |        |        |
> |-------|-----------|:--------:|:------:|:------:|:------:|:--------:|:------:|:------:|:------:|
> |       |           | P↑       | R↑     | F1↑    | AssA↑  | P↑       | R↑     | F1↑    | AssA↑
> | **Proprietary**  | Claude-3.5-Haiku   | 0.415    | 0.371    | 0.373    | 0.243    | 0.435      | 0.353      | 0.373      | 0.245      |
> |   | Claude-3.7-Sonnet  | 0.511    | 0.427    | 0.450    | 0.306    | 0.517      | 0.408      | 0.439      | 0.302      |
> |   | Gemini-1.5-Pro     | 0.471    | 0.470    | 0.454    | 0.309    | 0.477      | 0.454      | 0.451      | 0.309      |
> |   | Gemini-2.0-Flash   | 0.472    | 0.472    | 0.456    | 0.311    | 0.489      | 0.456      | 0.458      | 0.315      |
> |   | Gemini-2.5-Pro     | 0.473    | 0.586    | 0.507    | 0.355    | 0.504      | 0.575      | 0.525      | 0.374      |
> |   | GPT-4o-mini        | 0.167    | 0.156    | 0.154    | 0.087    | 0.178      | 0.149      | 0.155      | 0.088      |
> |   | GPT-4o             | 0.507    | 0.441    | 0.458    | 0.315    | 0.516      | 0.421      | 0.451      | 0.316      |
> | **Open-source**  | InternVL3-9B       | 0.290    | 0.302    | 0.284    | 0.174    | 0.301      | 0.296      | 0.288      | 0.179      |
> |  | InternVL3-14B      | 0.432    | 0.374    | 0.391    | 0.258    | 0.438      | 0.363      | 0.387      | 0.257      |
> |  | InternVL2.5-26B    | 0.386    | 0.383    | 0.369    | 0.239    | 0.392      | 0.372      | 0.367      | 0.240      |
> |  | Qwen-VL-2.5-7B     | 0.251    | 0.306    | 0.258    | 0.156    | 0.256      | 0.300      | 0.260      | 0.159      |
> |  | Qwen-VL-2.5-32B    | 0.427    | 0.350    | 0.375    | 0.245    | 0.420      | 0.331      | 0.360      | 0.236      |
>
> Please let us know if you have any further technical questions or concerns about our method and results and we would be happy to discuss them! Thanks again!

---

> ### Comment · Reviewer_BBAE · 2025-08-06
> **Re: Attention-based MLSM result**
>
> I think comparing MLSM with and without gaze weighting can indeed paint a picture on how well the current models prioritize objects that are more meaningful to humans. And from the results you show, it does indeed look like most models do not work better on more salient objects (with the exception of, for instance Gemini-2.5-Pro). I am just not sure why the conclusions are being made based on how well the model ranking is preserved, as opposed to just looking at individual models, comparing their results in, for instance, MLSM and HMLSM, and pointing out which change and which don't (since the ranking can in principle also be preserved in the case where all the models are equally good at prioritizing important objects). Other than that, I think this is indeed an interesting and important finding, I am just not sure about that piece of reasoning.

---

> > ### Author Response · Authors · 2025-08-06
> >
> > Thanks for your insightful comment and feedback. We think the problems you raised are quite important to strengthen our paper.
> >
> > **MLSM and H-MLSM.** We definitely agree with you that a separate comparing of the results for each model will provide a more detailed illustration. The reason we draw this conclusion based on the general ranking of MLLMs is that in the training of most existing MLLMs, the inclusion of spatial information is either depth or 3D bounding box [1], [2]. Since human attention has not explicitly fed in MLLMs during training, we can only expect an implicit representation in their output. Thus, considering the difference between relative performance ranking on general evaluation metrics such as MLSM, SRGS and SRD and attention-based (human-like) evaluation metrics can indeed provide some insights about the whether MLLMs can perform some extent of human-like understanding of the scene. We will look at the results of each MLLM individually and provide some discussion about it in our final version of paper.
> >
> > **How Sem is calculated.** The calculation of Sem we use is:
> > $\frac{\text{Unbiased Standard Deviation}}{\sqrt{n}}$
> > where $n$ is the number of re-runned samples. For example, for MLSM-P in GPT-4o, it is $\frac{0.172}{\sqrt{90}} = 0.018$.
> >
> > [1]: Boyuan Chen, Zhuo Xu, Sean Kirmani, Brain Ichter, Dorsa Sadigh, Leonidas Guibas, Fei Xia. SpatialVLM: Endowing Vision-Language Models with Spatial Reasoning Capabilities. CVPR 2024 \
> > [2]: An-Chieh Cheng, Hongxu Yin, Yang Fu, Qiushan Guo, Ruihan Yang, Jan Kautz, Xiaolong Wang, Sifei Liu. SpatialRGPT: Grounded Spatial Reasoning in Vision-Language Models. NIPS 2024
> >
> > Thanks again for your efforts in reviewing this paper! Please let us know if you have any further technical questions or concerns about our method and results and we would be happy to discuss them!

---

### Official Review · Reviewer_zSnK · 2025-07-01

**Clarity:** 4
**Significance:** 3
**Originality:** 3
**Rating:** 5
**Confidence:** 4

**Summary:**

The present study is concerned with representing the spatial locations and relationships among static and dynamic
objects in Autonomous Driving (AD) environments. In the proposed graph-based representation, objects are nodes loosely
located in a bird's eye view (BEV) regular grid. Relationships are encoded as edges corresponding to proximity and
ego-vehicle centric directional relations (e.g. "car A is at the back of car B"). Furthermore, the authors assign
visual saliency to objects by placing attention weights into the grid cells.

The proposed representation, termed SIG (Spatial Intelligence Graph) is employed both as ground truth and as model
output:
  - **ground truth** - the authors introduce a benchmark dataset (SIGBench), consisting of approximately
1400 frames annotated with three elements:
    - Spatial Intelligence Graphs (SIGs)
    - Spatial Relationship Paragraphs (SRPs) - a textual representation of the relationships in the SIGs
    - human gaze annotations (predicted with off-the-shelf methods and back-projected from image space onto the SIGs BEV grid)
  - **model output** - the authors prompt state-of-the-art Multimodal Large Language Models (MLMMs) with:
    - road scenes, asking to predict SIGs
    - road scenes, asking to perform SRP preposition completion
    - a temporal history of frames and saliency maps, with instructions to predict the visual saliency grid values for
the current frame.

Three metrics are used:
 - Multi-Level Spatial Matching: to capture the qualitative object localization/classification
 - Spatial Relationship Graph Similarity: to capture overall graph structure
 - Semantic Relational Distance: to capture the fidelity of particular inter-object proximal/directional relations

The authors conduct experiments involving both open-source and proprietary MLLMs. They mainly find that:
 - the selected proprietary MLLMs out-perform the open-source MLLMs on SIG an SRP prediction
 - spatial intelligence can be significantly improved by in-context learning with a random set of (image, SIG pairs) as
prompts
 - MLLMs cannot effectively predict human visual saliency in SIG space

**Questions:**

- Does the SIGBench dataset also preserve object IDs? Would it allow for tracking dynamic objects across individual
SIG frames?
- The model and benchmark seem to rely on single-camera and single frame information. This makes depth prediction
ambiguous up to scale. Have the authors also experimented with scale-invariant matching against the BEV locations?
- The grid resolution seems to be critical for the evaluation metrics. How do the authors decide on the value to
use?

**Ethical Concerns:**

["NO or VERY MINOR ethics concerns only"]

**Final Justification:**

I thank they authors for their detailed responses. Their rebuttal has addressed my concerns. In particular, I think the inclusion of lane markings (albeit on a subset of the dataset) and the generalization to other domains would greatly strengthen the contribution of the present work.

In future manuscripts, I encourage the authors to include the choice of a single-frame setup as a limitation, and to discuss their analysis on the grid size and the saliency map templating strategy in the appendix.

**Limitations:**

This reviewer thinks the authors should point out that the method:
 - relies on predicted human visual saliency, hence suffers from systematic prediction errors induced by the respective methods.
 - does not consider lanes and their relationships to the objects in the driving environment, which may
 limit applicability for AD

**Quality:**

3

**Strengths And Weaknesses:**

Strengths:
  - Well written paper and clearly structured work
  - High importance, as spatial scene understanding is an outstanding issue in MLLMs applied to AD, especially with
the emergence of end-to-end driving
  - Extensive evaluation

Weaknesses:

 - There is no human baseline for the proposed metrics (e.g. based on consistency between human subjects). It is
difficult to judge how far the SOTA spatial intelligence level really is compared to the gold standard of human drivers
 - The study omits lanes from the evaluation (lines 130-131). However, proper understanding of the lane and drivable
space structure are crucial for AD. Furthermore, by omitting lanes, the study misses on the opportunity to model
higher level driving scene spatial layouts, such as "vehicle 1 on the left lane" or "traffic sign 1 corresponds to the
ego lane". Such relationship are of primary importance for AD (consider overtaking or stopping for a red light).
 - Dynamic spatial relationships between objects (e.g. vehicles approaching a pedestrian crossing or accelerating
towards the ego vehicle) are not covered
 - The BEV grid resolution is somewhat coarse (10x10), and the authors do not provide an intuition on how to calibrate
it
 - The saliency prediction prompt seems to ask the MLLM to generate python code for producing the attention map. It is
not clear to what extent this setup affects the outcome.

---

> ### Author Rebuttal · Authors · 2025-07-29
>
> We want to thank the reviewer for your efforts in handling this submission and we appreciate you acknowledging the importance of our work in improving spatial scene understanding of MLLMs in autonomous driving (AD) scenario, along with extensive evaluation. We will address your questions one by one in the following.
>
> **W-1 No human baseline.** We agree that a human baseline is essential for contextualizing model performance. Accordingly, in Tab. 2 (first row) we report the average scores of four human subjects on our general VSI tasks. We will emphasize these results more prominently in the revised experiments section to clarify how current SOTA models compare against human-level spatial reasoning.
>
> **W-2&L-2  Omitting lanes from the evaluation.** We agree with the importance of lane and drivable‐space understanding in AD scenarios. You are absolutely right that lane assignments (e.g. “Car 1 is in the left lane”) is critical for maneuver planning. Our SIGBench dataset does include human‐annotated lane masks for images where lanes are visible. Currently, there are several mature methodologies specifically designed for traffic lane detection such as [1], [2], [3]. However, in this work we mainly focus on the spatial relationship among objects rather than spatial traffic rules. Moreover, SIGBench encompasses a diverse range of driving scenarios: some featuring faded or blurred lane markings and others (e.g., intersections or rural roads) where lane demarcations are entirely absent. In the revised manuscript, we will select a subset of images with clearly visible lane annotations and extend our evaluation metrics to incorporate lane‑based spatial relationships for this subset.
>
> **W-3 Dynamic spatial relationships between objects.** Thank you for this insightful suggestion. We agree that modeling dynamic spatial relationships, such as vehicles approaching a pedestrian crossing or accelerating toward the ego vehicle, is a critical aspect of visual‑spatial intelligence. As SIGBench currently focuses on single‑image scenarios, inferring motion or intent from a static frame can be challenging even for human observers. However, because our source data consists of video sequences, we intend to revisit these clips and introduce temporal annotations in a future extension of the benchmark. This enhancement will allow evaluation of models’ ability to reason about object trajectories and evolving spatial configurations.
>
> **W-4&Q-3 The 10x10 grid resolution is coarse.** Choosing 10×10 discretization for our Spatial Intelligence Grid (SIG) was motivated by typical U.S. highway layouts: mostly up to four lanes per direction (eight total), plus one extra column on each side for signs and lights, yielding 1+8+1=10 horizontal cells. To form a square grid, we also use 10 rows vertically. \
> To give an intuition for how this maps to real‑world dimensions, consider a reference plane at 27m in front of the camera. The camera used for recording has focal length 3.6mm and active area =5.4mmx4.0mm. Since our image size is $1920\times1080$, we can get the focal length in pixels as:
> $$f_x = \frac{3.6mm}{5.4mm} \times 1920 = 1280\text{px}, \quad f_y = \frac{3.6mm}{4.0mm} \times 1080 = 972\text{px} $$
> If we divide the image into 10x10 grids, each image cell is 192px wide ($\Delta u$) and 108px tall ($\Delta v$). If we have $Z = 26$m, we can solve the width ($\Delta X_{grid}$) and height ($\Delta Y_{grid}$) of each cell in our SIG by:
> $$
>     \Delta X_{grid} = \frac{\Delta u}{f_x}Z = \frac{192\text{px}}{1280\text{px}}\times 27m \approx 4m, \quad \Delta Y_{grid} = \frac{\Delta v}{f_y}Z = \frac{108\text{px}}{972\text{px}}\times 27m \approx 3m
> $$
> which means each grid in our SIG represents approximately a 4mx3m rectangle plane in real world. We also provide more comprehensive results for SIG-based ICL, please refer to reviewer **BBAE W-2 and Q-4**. Our SIG representation can also be generalized to other domains besides AD, please refer to reviewer **wMQv W-1&Q-2**.
>
> **W-5 Whether asking MLLM to generate python code for saliency prediction affects the outcome.** We choose a Python‑code prompt for saliency prediction generation using MLLMs for two main reasons:
> - Token‑limit constraints: A full 1080x1920 attention map contains 2073600 values, far exceeding the maximum output capacity of our evaluated models (e.g., 64000 tokens for GPT‑4o, which is already much larger than that of most proprietary and open-source models). If we want the GPT-4o to output the full attention map of one image, we need 2073600/64000 = 32.4 iterations. Thus, asking the model to emit code that reconstructs the map circumvents this limit by having the model produce a compact algorithm rather than raw data, which can then be executed to yield the full attention tensor.
> - Execution robustness: By prescribing a code template that references only our approved libraries and data structures, we ensure that the generated routines run correctly and consistently in our environment. This approach also allows us to validate and debug any edge cases in advance, increasing overall reliability.
>
> Through extensive prompt engineering and manually checked over 100 cases, we found that this code‑templating strategy does not bias the resulting attention maps compared to raw‑value generation. Instead, it simply provides a practical mechanism to deliver high‑resolution saliency outputs within existing MLLM constraints.
>
> **Q-1 Does the SIGBench dataset also preserve object IDs that supports tracking?** While SIG provides a natural schema for object identity and could facilitate cross‑frame tracking, SIGBench in its current form is limited to single‑image annotations and does not include persistent object IDs. Establishing reliable object tracks would require extensive video‑based labeling, which is beyond the scope of this initial release. We agree that integrating object IDs and temporal relationships would greatly enhance the benchmark, and we plan to explore this extension along with the dynamic object relationship in a future SIG video dataset.
>
> **Q-2  Scale-invariant matching.** We agree with you that relying on a single camera and frame can introduce scale ambiguity in depth estimation. Our choice of this setting was deliberate: by demonstrating that SIG‑based ICL improves visual‑spatial reasoning under these challenging constraints, we establish a strong foundation. Importantly, our SIG representation is agnostic to sensor modality or scale: it can be directly extended to multi‑camera rigs or LiDAR–camera fusion systems that yield metrically accurate depth.\
> Although we have not yet performed explicit scale‑invariant matching against BEV ground truth, our grid‑based metrics: Multi-level Spatial Matching (MLSM) and Spatial Relation Graph Similarity (SRGS), operate on normalized cell indices rather than absolute distances, making them naturally robust to uniform scale shifts. More specifically:
> - MLSM builds a cost matrix based on the cell‑coordinate differences between predicted and ground‑truth objects; since both are expressed in the same 10×10 index space, any uniform scaling of the scene (e.g., due to camera zoom or depth ambiguity) cancels out.
> - SRGS compares the structure of scene graphs via node and edge matches. Its graph‑edit operations count mismatches in graph topology and relation labels rather than measuring literal distances in meters.
>
> As a result, both metrics remain unaffected by global scale shifts: they focus purely on relative spatial configuration and object relationships, ensuring consistent evaluation across different calibration settings.
>
> **L-1 Predicted human visual saliency will result in systematic prediction error.** We apologize for any misunderstanding. In our experiments, ground‑truth human visual saliency was captured using the SmartEye Pro 5 eye‑tracking system. According to the device specifications and validation studies conducted by SmartEye, the gaze‑direction measurements achieve a precision of approximately $\pm0.5^{\circ}$ [4]. This level of accuracy is sufficient for our purposes, and we therefore consider these recordings to represent reliable ground‑truth saliency.
>
> [1] Xingang Pan, Jianping Shi, ..., Xiaoou Tang. SCNN: Spatial As Deep: A Convolutional Neural Network for Scene Segmentation of Lane Markings. ECCV 2018 \
> [2] Lucas Tabelini, Rodrigo Berriel1, ..., Thiago Oliveira-Santos. Keep your Eyes on the Lane: Real-time Attention-guided Lane Detection. CVPR 2021 \
> [3] Lizhe Liu, Xiaohao Chen, Siyu Zhu, Ping Tan. CondLaneNet: a Top-to-down Lane Detection Framework Based on Conditional Convolution.  ICCV 2021\
> [4] Smart Eye AB. SmartEye Pro 5 User Manual (Revision 249)

---

> > ### Comment · Reviewer_zSnK · 2025-08-07
> >
> > I thank the authors for their detailed response. Their rebuttal has clarified my questions regarding the human baselines
> > both in terms of scene understanding, gaze ground truth and the methodology used to select the grid size. My concerns
> > regarding lane markings are also addressed by the proposed evaluations using lane detection methods on images on which
> > lane marking are visible.
> >
> > Related to the dynamic spatial relationships, scale-invariant matching and object id tracking, it seems we are in
> > agreement that moving beyond single-frame setups, although costly, would enhance the benchmark. I would encourage the
> > authors to mention that single-frame inference and labeling is a limitation of the present work.
> >
> > Finally, regarding the python code generation approach to saliency, I understand the motivation for using python
> > code as a proxy representation. However, I would be curious to find out what methodology was used to rule out the
> > induced biases (i.e. the actual criteria used in the manual qualitative check). Please also include some details on this
> > in the final manuscript.

---

> ### Author Response · Authors · 2025-08-03
> **Update the complete results of SIG-COCO and SIG-ARKitScenes and results of new data sample selection strategy for SIG-based ICL**
>
> Please refer to our new comment to reviewer **wMQv W-1&Q-2** for the udpated results on SIG-COCO and SIG-ARKitScenes, which demonstrates the strong generalizability of our proposed SIG to domains other than autonomous driving scenario.
>
> Please refer to our new comment to reviewer **BBAE W-2** for the new results on SIG-based ICL with another data sample selection strategy, which highlights the potential of SIG as a new data schema for improving VSI of MLLMs.

---

> > ### Author Response · Authors · 2025-08-06
> > **Happy to hear from you and discuss**
> >
> > Dear Reviewer zSnK,
> >
> > We appreciate you bringing the idea of dynamic spatial relationship between objects and scale-invariant matching to our attention and it is helpful! Your feedback provides two novel directions of our future work!
> >
> > For the inclusion of traffic lanes, we are working on selecting a subset that contains clearly and visible traffics lanes and designing evaluation metrics that includes traffic lanes. For the SIG-based ICL, you can refer to our latest comment to reviewer **BBAE** for the comparison of multi-run error bar statistics between text(multiple-choice vqa)-based ICL and SIG-based ICL on both GPT-4o and Gemini-2.5-Pro, which again highlights the potential of SIG as a new data representation for improving MLLMs' visual-spatial intelligence (VSI).
> >
> > Also, to better support future works for other researchers, we have confirmed with U.S.DOT and decided to release the SIGBench and SIGBench-tiny dataset, including original driving scenes images, images annotated with bounding boxes of vehicles, human-annotated SIG, SRP template with answers and human attention for each image. We believe our benchmark will provide a new persecptive in assessing the VSI of MLLMs and the first benchmark to evaluate it including human-like attention.
> >
> > We look forward to hearing your feedback and, we will be more than happy learn from your comments and thoughts!
> >
> > Sincerely,
> > Authors

---

> ### Author Response · Authors · 2025-08-07
>
> Thank you for your appreciation of our rebuttal. Your thoughtful feedback has clearly strengthen our paper.
>
> Your ideas about dynamic spatial relationships between objects, scale-invariant matching and object id tracking are insightful and represent promising topics for our future work. We will explicitly discuss these aspects as limitations of our current work and directions for future work in the final version of our paper.
>
> Regarding the Python code generation for gaze prediction, our key idea is ensuring the executability of the generated code. For instance, if the model outputs code that imports libraries not available in our conda environment, it fails to produce the predicted gaze attention and instead raises runtime errors. To mitigate this, we explicitly constrain the allowed libraries within the prompt and will automatically rerun the inference until the code is executable. Additionally, during manual inspection, we focus on whether the MLLMs are truly performing gaze prediction based on the attention from previous five frames, rather than simply averaging them. We have reported such occurrences in the manuscript (Lines 284–285) and provided further explanation in Appendix C.3. We will include more detailed clarification on this in the final version.
>
> Thank you once again for your valuable efforts in reviewing our work. Should you have any further technical questions or suggestions, we would be glad to address them and continue the discussion.

---

### Official Review · Reviewer_wr1L · 2025-07-01

**Clarity:** 3
**Significance:** 3
**Originality:** 4
**Rating:** 5
**Confidence:** 3

**Summary:**

This paper proposes a novel grid-based representation, Spatial Intelligence Grid (SIG), for encoding Visual-Spatial Intelligence (VSI) in multimodal large language models (MLLMs), particularly in the context of autonomous driving (AD). The authors argue that existing Visual Question Answering (VQA)-style representations inadequately capture spatial reasoning.
They introduce SIG, a structured grid-based scene representation that embeds spatial and physical priors, along with a new benchmark, SIGBench, comprising over 1,400 annotated driving frames with human gaze data and ground-truth SIG labels.
Extensive experiments on several state-of-the-art MLLMs demonstrate that SIG-based few-shot in-context learning (ICL) improves performance in spatial reasoning tasks compared to traditional VQA-style prompts. Additionally, SIG enables human-like VSI assessment through attention-modulated metrics.

**Questions:**

- How well would SIG generalize to domains beyond autonomous driving? Could the authors clarify or expand on this?
- What is the runtime overhead of the proposed graph-based metrics, and are they practical for real-time applications like autonomous driving?
- Can the authors provide an analysis of common failure cases to better understand model limitations?

**Ethical Concerns:**

["NO or VERY MINOR ethics concerns only"]

**Final Justification:**

I appreciate the authors’ thoughtful rebuttal, particularly the additional analyses on runtime and failure cases, which have addressed my main concerns and strengthened the paper’s contributions. Given these improvements, I recommend Accept, under the condition that the additional analyses and clarifications provided in the rebuttal are incorporated into the camera-ready version. I trust the authors to ensure that these enhancements are faithfully integrated.

**Limitations:**

Yes

**Quality:**

3

**Strengths And Weaknesses:**

Strengths:
- The paper is well-written, clearly structured, and effectively presents its main contributions. The integration of figures with the textual explanations is useful, providing clarity and improving the reader’s understanding of the proposed method.
- SIGBench is thoughtfully designed for both grid-based and human-like VSI evaluation, incorporating gaze-based annotations and rigorous, graph-theoretic evaluation metrics. The proposed metrics (MLSM, SRGS, and SRD) are particularly well-motivated, offering a structured and interpretable means of quantifying spatial reasoning performance beyond simple right-wrong QA scoring. They allow for fine-grained assessment of relational and positional errors, which is especially valuable in complex environments like autonomous driving.
- The authors extensively benchmark multiple proprietary and open-source MLLMs, providing strong empirical evidence that SIG consistently improves spatial reasoning in both zero-shot and in-context learning setups.
- The discovery that current MLLMs fail to appropriately prioritize objects based on attentional salience, as humans naturally do in driving tasks, is a valuable insight. Moreover, the choice of autonomous driving as a testbed is appropriate given its real-world complexity and practical relevance.

Weaknesses:
- While autonomous driving is an appropriate and demanding testbed, the paper does not explore how SIG might generalize to other domains. While this aspect is being discussed briefly in the Appendix I think it still is worth noting.
- The paper introduces graph-edit distance and semantic relational distance metrics, but it does not address their computational costs during large-scale inference. Given that the target application is autonomous driving, where real-time efficiency is crucial, an analysis of runtime performance of the pipeline would be valuable.
- The paper would benefit from a qualitative or quantitative analysis of typical failure modes, such as common errors in SRGS predictions or attention misallocations. This could help identify systematic weaknesses in current models and inform future improvements.
- The primary experimental advantage demonstrated stems from in-context learning (ICL) with SIG, while no experiments on full fine-tuning or reinforcement learning with human feedback (RLHF) are presented. This leaves open the important question of how effectively SIG could be integrated into end-to-end training pipelines for production systems.

Other issues not affecting the rating:
- Typo at line 120
- Line 254 -> Pearson

---

> ### Author Rebuttal · Authors · 2025-07-29
>
> We want to thank the reviewer for your efforts in handling this submission and we appreciate you acknowledging our proposed evaluation metrics are well-motivated and our extensive experiments provides strong empirical evidence for Spatial Intelligence Grid (SIG)'s potential in improving visual-spatial intelligence (VSI) of MLLMs. We will address your questions one by one in the following.
>
> **W-1&Q-1 Can SIG generalize to other domains than AD.** Please refer to our response to reviewer  **wMQv W-1&Q-2**
>
> **W-2&Q-2 Analysis of runtime.** Runtime efficiency is definitely important, especially for real‑time applications in autonomous driving (AD). Below we summarize both the theoretical complexity of our metrics and empirical timing results on SIGBench. \
> Theoretical time complexity: Let $n$ denotes the total number of objects (vehicles, traffic signs and traffic lights) per frame.
>   - For Multi-level Spatial Matching (MLSM), the time complexity for building cost matrix for vehicles, traffic signs and traffic lights is $O(n^2)$. The time complexity for bipartite matching is $O(n^3)$ [1]. For multi-threshold matching, the time complexity is $O(n)$. The overall time complexity for MLSM is $T_{MLSM}(n) = O(n^2) + O(n^3) + O(n) \Rightarrow O(n^3)$.
>   - Similarly, for Spatial Relation Graph Similarity (SRGS), during the fully connected graph construction, we require $O(n)$ for all nodes and $O(n^2)$ for all edges. Next, the bipartite matching requires $O(n^3)$ similarly. During the graph edit distance (GED), we are performing node-to-node and edge-to-edge comparison, which requires $O(n) + O(n^2)$. Thus, the time complexity for SRGS is $T_{SRGS} = 2(O(n) + O(n^2)) + O(n^3) \Rightarrow O(n^3)$.
>   - For Semantic Relational Distance (SRD), we are computing pointwise distance. For SRD (Directional) and SRD (Proximal) there are 8 and 5 possible relations, respectively. Thus, the time complexity for SRD is $T_{SRD} = O(8n) + O(5n) \Rightarrow O(n)$.
>
> Empirical timing: The dataset statistics of SIGBench is:
> | #Objects | 2   | 3   | 4   | 5   | 6   | 7   | 8   | 9   | 10  | 11  | 12  | 13  | 14  | 15  | 16  | 17  | 18  | 19  | 20  | 21  | 22  |
> |----------|-----|-----|-----|-----|-----|-----|-----|-----|-----|-----|-----|-----|-----|-----|-----|-----|-----|-----|-----|-----|-----|
> | #Samples | 145 | 204 | 219 | 180 | 145 | 105 | 99  | 59  | 47  | 38  | 33  | 18  | 15  | 6   | 6   | 3   | 0   | 2   | 0   | 0   | 1   |
>
> The average per-frame runtimes on CPU (AMD Ryzen 5900X) are:
> | #Objects  | 3      | 5      | 7      | 9      | 11     | 13     | 15     | 17     | 19     | 22     |
> |-----------|--------|--------|--------|--------|--------|--------|--------|--------|--------|--------|
> | MLSM [s]  | 0.0001 | 0.0002 | 0.0002 | 0.0002 | 0.0002 | 0.0002 | 0.0002 | 0.0003 | 0.0003 | 0.0004 |
> | SRGS [s]  | 0.0001 | 0.0002 | 0.0003 | 0.0004 | 0.0004 | 0.0005 | 0.0006 | 0.0006 | 0.0008 | 0.0009 |
> | SRD [s]   | <0.0001| <0.0001| <0.0001| <0.0001| <0.0001| <0.0001| <0.0001| <0.0001| <0.0001| <0.0001|
>
> Estimated real-time performance: Consider a typical AD vehicle with six cameras at 30 FPS and an average of 11 detected objects per frame. The total estimated time cost for our metrics per second (without parallelization) is: $6 \times 30 \times (0.0002 + 0.0004 + 0.0001 ) = 0.126$s, which definitely meets the real‑time demands. We will include these discussion in the Appendix.
>
> **W-3&Q-3 Qualitative and quantitative analysis of typical failure modes.** To explore common failure modes, we examine zero‑shot performance of two state‑of‑the‑art MLLMs on SIGBench across scenes containing varying numbers of objects. Results are summarized in the following table.
>
> | Type               | #Objects |**MLSM**     |    |        |         | **SRGS** |       | **SRD (Directional)** |    |          |  **SRD (Proximal)**  |      |          |
> |--------------------|:--------:|:--------:|:------:|:------:|:-------:|:--------:|:-----:|:---------------------:|:------:|:--------:|:--------------------:|:------:|:--------:|
> |                    |          | P↑       | R↑     | F1↑    | AssA↑   | S↑       | WS↑   | MAE↓                  | MSE↓   | Acc↑     | MAE↓                 | MSE↓   | Acc↑     |
> | **GPT‑4o**         | 3        | 0.541    | 0.525  | 0.521  | 0.371   | 0.425    | 0.422 | 1.951                 | 5.386  | 0.152    | 0.760                | 1.061  | 0.386    |
> |                    | 7        | 0.473    | 0.346  | 0.394  | 0.255   | 0.249    | 0.238 | 1.967                 | 5.415  | 0.138    | 1.043                | 1.789  | 0.271    |
> |                    | 11       | 0.460    | 0.291  | 0.354  | 0.219   | 0.203    | 0.186 | 1.027                 | 1.602  | 0.245    | 0.383                | 0.457  | 0.654    |
> |                    | 16       | 0.429    | 0.219  | 0.288  | 0.171   | 0.114    | 0.099 | 0.433                 | 0.553  | 0.627    | 0.080                | 0.080  | 0.920    |
> |                    | 22       | 0.465    | 0.190  | 0.270  | 0.160   | 0.098    | 0.132 | 1.520                 | 3.280  | 0.200    | 0.440                | 0.440  | 0.560    |
> | **Gemini‑2.5‑Pro** | 3        | 0.482    | 0.643  | 0.538  | 0.384   | 0.212    | 0.232 | 1.219                 | 2.582  | 0.270    | 0.755                | 1.186  | 0.431    |
> |                    | 7        | 0.492    | 0.547  | 0.510  | 0.353   | 0.228    | 0.213 | 1.321                 | 2.932  | 0.253    | 0.798                | 1.216  | 0.387    |
> |                    | 11       | 0.447    | 0.503  | 0.468  | 0.311   | 0.226    | 0.181 | 0.700                 | 1.068  | 0.441    | 0.318                | 0.381  | 0.713    |
> |                    | 16       | 0.544    | 0.487  | 0.511  | 0.351   | 0.287    | 0.252 | 0.480                 | 0.667  | 0.613    | 0.060                | 0.060  | 0.940    |
> |                    | 22       | 0.565    | 0.437  | 0.493  | 0.327   | 0.407    | 0.357 | 1.000                 | 1.960  | 0.360    | 0.760                | 1.160  | 0.440    |
>
> GPT‑4o’s MLSM and SRGS scores steadily decline as object count rises, while its SRD performance fluctuates, demonstrating that graph‑based metrics are more sensitive to scene density and that raw text judgments alone can mask these trends. Gemini‑2.5‑Pro shows similar SRD variability but even improves on some graph metrics (MLSM‑P, SRGS) with more objects, underscoring the complementarity of SIG’s structured representation.
> Qualitatively, we recognize two key failure scenarios:
> - Objects that are either very small or positioned at the periphery of the scene are frequently missed or mislocalized due to their low visual salience.
> - High intersection over union (IoU) between objects often cause the models to conflate object identities or assign them to incorrect grid cells, resulting in swapped or merged scene‑graph nodes. We will illustrate each of these failure modes with representative examples with visualization in the Appendix.
>
> **W-4 Full fine-tuning or RLHF based on SIG.** Thank you for this insightful suggestion. We agree that exploring end‑to‑end integration of SIG into training pipelines could substantially advance model capabilities. We provide more comprehensive results for SIG-based ICL, please refer to reviewer **BBAE W-2 and Q-4**. Based on our positive ICL results, we think two particularly promising directions are:
> - SIG‑guided supervised fine‑tuning. Rather than relying solely on in‑context examples, we could curate a labeled SIG dataset and fine‑tune an MLLM to predict grids and scene graphs directly from raw images. This supervised signal would reinforce precise spatial encoding, potentially yielding improvements not only in SIG reconstruction but also in downstream tasks like VQA or navigation with an improved visual-spatial intelligence with large generalizability.
> - RLHF via SIG. SIG outputs offer a structured, graph‑based explanation of model reasoning. By treating human judgments of SIG quality as reward feedback, e.g., ranking or correcting scene graphs, an RLHF loop could encourage models to align their spatial representations with human preferences. Such a protocol may cultivate nuanced attention patterns and more robust relational reasoning tailored to diverse driving contexts. Both avenues harness the rich structure of SIG to move beyond one‑shot prompting toward production‑ready spatial intelligence, and we will investigate these two interesting directions in future work.
>
> **Typo at line 120; Line 254 $\rightarrow$ Pearson.** We are deeply apologize for these typos and we will correct them in the final version of this paper.
>
> [1] T. H. Cormen, C. E. Leiserson, R. L. Rivest, C. Stein, Introduction to Algorithms, 3rd ed., MIT Press, 2009, Chapter 16.

---

> ### Author Response · Authors · 2025-08-03
> **Update the complete results of SIG-COCO and SIG-ARKitScenes and results of new data sample selection strategy for SIG-based ICL**
>
> **W-1&Q-1 Can SIG generalize to other domains than AD.** Please refer to our new comment to reviewer **wMQv W-1&Q-2** for the udpated results on SIG-COCO and SIG-ARKitScenes.
>
> Please refer to our new comment to reviewer **BBAE W-2** for the new results on SIG-based ICL with another data sample selection strategy, which highlights the potential of SIG as a new data schema for improving VSI of MLLMs.

---

> > ### Comment · Reviewer_wr1L · 2025-08-05
> >
> > I appreciate the authors’ detailed rebuttal, particularly the additional analyses on runtime and failure cases, as well as the discussion on integrating the proposed approach into an end-to-end pipeline. These additions directly address my earlier concerns and significantly strengthen the paper’s contributions.

---

> > > ### Author Response · Authors · 2025-08-05
> > >
> > > Thank you so much for your comments and expertise! Your review and insights have clearly strengthened our work. If you have any further questions or ideas, we would be more than happy to discuss!
> > >
> > > Authors

---

> > > ### Author Response · Authors · 2025-08-07
> > > **Thank you!**
> > >
> > > Thanks again for your efforts in reviewing this paper. Your insightful feedback and suggestions strengthen our paper a lot and provides new perspective that we have not considered before. We will definitely include the content of our discussion into our final version of paper!

---

### Official Review · Reviewer_wMQv · 2025-07-02

**Clarity:** 2
**Significance:** 3
**Originality:** 3
**Rating:** 3
**Confidence:** 4

**Summary:**

The paper proposes a novel approach and benchmark for evaluation of visual spatial intelligence of MLLMs.
The authors introduce spatial grid based methods  for assessing spatial relationships in Autonomous Driving scenarios and evaluate whether current MLLMs  can answer spatial queries, generate the spatial grid from images using  in-context learning. The gaze of human differers is examined in comparison with  machine based approaches.

**Questions:**

- Provide examples of queries and how they differ for the queries in other benchmarks such as SpatialVLM or VSR benchmarks and how they are obtained for SIG
-  How does the approach extend to other scenarios beyond Autonomous Driving

**Ethical Concerns:**

["NO or VERY MINOR ethics concerns only"]

**Final Justification:**

The rebuttal and the discussion improved my appreciation of the results leading to the increase of my rating. I still have some outstanding concerns about insights to MLMM's alone and the representations that are learned in the pre-training stage.

**Limitations:**

Yes.

**Paper Formatting Concerns:**

None.

**Quality:**

2

**Strengths And Weaknesses:**

Strengths:
- novel benchmark for map style representation for assessing spatial relationships
- novel evaluation metric, using spatial relation graph similarity, multi-level spatial matching


Weakness
- this is likely applicable only Autonomous driving scenarios
- the positioning of the work with respect to related works could be improved,
- the method is effectively a map based methods, see [1] for additional references
- better motivation for new evaluation metrics should be provided
- the connection to human gaze evaluation is weak

The paper is not well organized. The discussion the evaluation metrics precedes the discussion of grid computation and the benchmark, there are very few examples of prompts, so it is hard to address the complexity, strengths and weaknesses of the method. The approach suggests to use in context learning  to obtain the spatial grid json format, which is now well aligned with the benchmarks where the models reason
about spatial relations in Bird Eye View map.

Please look at this paper and references within,
[1] https://arxiv.org/pdf/2410.06468
DOES SPATIAL COGNITION EMERGE IN FRONTIER
MODELS? Santhosh Kumar Ramakrishnan∗ Erik Wijmans Philipp Krahenb ¨ uhl Vladlen Koltun

---

> ### Author Rebuttal · Authors · 2025-07-29
>
> We want to thank the reviewer for your efforts in handling this submission and we appreciate you acknowledging the novelty of our proposed data representation Spatial Intelligence Grid (SIG) and evaluation metrics: Multi-level Spatial Matching (MLSM), Spatial Relation Graph Similarity (SRGS) and Semantic Relational Distance (SRD). We will address your questions one by one in the following.
>
> **W-1&Q-2 This is likely applicable only for AD scenarios.** Although our initial motivation arise from autonomous‑driving (AD) applications, the SIG framework is fundamentally domain‑agnostic and can be applied wherever a fixed ontology of object types exists. To demonstrate this, we have built two proof‑of‑concept benchmarks based on subsets from MS COCO [1] and ARKitScenes [2], named as SIG-COCO and SIG-ARKitScenes, respectively. We run both zero-shot inference and ICL using GPT-4o on these two benchmarks. The results are shown in the following table. Due to time limitations and workload of dataset annotation, we only provide partial results of MLSM and SRGS here, we will update the results of SRD related metrics in 3-4 days.
>
> | Benchmark           | Type    | **MLSM** |        |        |        | **SRGS** |       |
> |---------------------|---------|:--------:|:------:|:------:|:------:|:--------:|:-----:|
> |                     |         | P↑       | R↑     | F1↑    | AssA↑  | S↑       | WS↑   |
> | SIG-COCO        | Z-S     | 0.758    | 0.826  | 0.771  | 0.652  | 0.574       | 0.607     |
> |                     | ICL-SIG | 0.860    | 0.840  | 0.849  | 0.749  | 0.711        | 0.746     |
> | SIG-ARKitScenes | Z-S     | 0.714    | 0.756  | 0.727  | 0.581  | 0.619        | 0.597     |
> |                     | ICL-SIG | 0.809    | 0.850  | 0.823  | 0.719  | 0.729       | 0.737     |
>
> **W-2 Position of this work with other related works.** Thanks for drawing our attention to the related literature. In particular, the paper [3] you suggested offers valuable insights closely aligned with our work. In the revised manuscript, we will integrate a discussion of [3] alongside citations to [4], [5], [6], and [7], clearly positioning our contributions in the context of these existing studies. If you are aware of any additional papers that would further enrich our comparison, we would be grateful for your recommendations and are happy to include them.
>
> **W-3 Similar as map-based method.** While the map‑based approach in [3] bears a conceptual resemblance to our SIG, our work departs in several important ways:
>
> - Scenarios from real‑world: We annotate SIG directly on real autonomous‑driving scenes rather than synthetic or abstract maps, ensuring our benchmarks measure performance under the complex visual conditions encountered “in the wild.”
> - Richer object classes: Beyond a generic “object” class, our SIG distinguishes multiple vehicle types (cars, trucks, vans, buses), traffic signs, and lights, and further encodes intra‑class attributes such as color and order.
> - Instance‑level complexity: By including scenes with several instances of the same category—each with its own spatial relation and attribute differences—we stress‑test a model’s ability to generalize spatial understanding to densely populated environments.
> - One‑shot relational output: Our evaluation framework invites MLLMs to output a complete SIG in a single pass, mirroring how humans naturally describe spatial relations in a cohesive narrative, rather than iterative queries of VQA in [3], which has been studied by enormous of existing papers.
>
> These distinctions underscore how SIG extends and challenges map‑based representations to better capture the visual‑spatial intelligence needed for real‑world driving applications.
>
> **W-4 Better motivation for proposed evaluation metrics.** When a driver is driving, he/she typically focus on a range containing selected objects and their spatial correlation instead of pixel-level or object-level information. To build a representation that preserve this, inspired from classical grid‑based visual schemes (e.g., Renaissance “drawing machines”) that artists used to decompose complex scenes into structured partitions, we introduce SIG that preserves fine-grained geometric and topological information in a 10×10 grid, directly encoding object positions and their spatial relations. Based on SIG, we design three complementary evaluation metrics:
>
> - MLSM: A thresholded, bipartite‑matching protocol that tolerates minor grounding noise while measuring localization precision at increasing levels of attribute matching (type, order, color).
> - SRGS: Based on graph edit distance (GED), this metric rigorously quantifies structural differences between predicted and ground‑truth scene graphs, penalizing missing or mis‐classified relations.
> - SRD: By mapping prepositions onto discrete scales (cyclical for direction, linear for proximity), SRD captures the degree of semantic error, rather than a binary correct/incorrect judgment.
>
> **W-5 Connection to human gaze evaluation is weak.** While the core contribution of this work lies in introducing SIG and SIG‑based evaluation metrics, modeling human‑like VSI via gaze serves two essential roles:
>
> - Bridge to real‑world decision making: In AD scenario, human drivers do not process every object equally. They deploy selective attention, focusing on a few critical elements such as traffic lights at an intersection—while deemphasizing others (e.g., distant vehicles). By integrating gaze‑derived saliency into SIG (Eq. 6), we weight our graph‑ and grid‑based metrics so that high‑importance objects incur greater penalty when mispredicted.
> - Human‑aligned relational evaluation: Traditional VQA or text‑only metrics treat each relation equally, ignoring that some spatial errors are more consequential than others. For instance, confusing “at the front of” versus “at the back of” a high‑attention object (e.g., a braking light) can have drastically different safety implications. Our Human‑Like SRGS and SRD scale node‑ and edge‑edit costs by gaze weights, ensuring that mistakes on visually salient entities are penalized more heavily.
>
>
> Together, these gaze‑weighted evaluations not only emulate how humans allocate attention in dynamic driving scenes but also align model assessment with the situational priorities of real‑world AD systems—thereby strengthening the practical relevance of our human‑like VSI benchmark.
>
> **Q-1 Examples of queries and difference from SpatialVLM and VSR.** For examples of queries, please refer to Appendix C. Below we illustrate how query formats in SpatialVLM, VSR, and SIGBench differ—both in their structure and in how they are obtained:
>
> | Benchmark  | Query Format                                                 | Evaluation Metrics                                           |
> | ---------- | ------------------------------------------------------------ | ------------------------------------------------------------ |
> | SpatialVLM | “How far is A from B (m)?”, “Is X larger than Y?”            | Mean absolute error (m), Size‑ordering accuracy              |
> | VSR        | “Object 1 is left of Object 2.” True/False?                  | True/False classification accuracy                           |
> | SIGBench (ours)   | General‑VSI: SIG creation in JSON format (SIGC) and spatial‑relational paragraph fill‑in (SRPF); Human‑like VSI: Gaze prediction + H‑SIGC + H‑SRPF | General‑VSI: MLSM, SRGS, SRD; Human‑like VSI: attention‑based SRGS/SRD; Gaze prediction: PCC, IG, KL‑Divergence |
>
> This direct, structured prompting—together with our ground‑truth SIG annotations—allows SIGBench to evaluate complete spatial understanding in one shot, rather than separate VQA pairs. We also provide more comprehensive results for SIG-based ICL, please refer to reviewer **BBAE W-2 and Q-4**.
>
> [1] Tsung-Yi Lin, Micheal Maire, ..., Piotr Dollar. Microsoft COCO: Common Objects in Context.\
> [2] Gilad Baruch, Zhuoyuan Chen, ..., Elad Schulman. ARKitScenes: A Diverse Real-World Dataset For 3D Indoor Scene Understanding Using Mobile RGB-D Data.\
> [3] Santhosh Kumar Ramakrishnan, Erik Wijmans, Philipp Krahenbuhl, Vladlen Koltun. DOES SPATIAL COGNITION EMERGE IN FRONTIER MODELS?\
> [4] Santhosh Kumar Ramakrishnan. Predictive scene representations for embodied visual search.\
> [5] Runsen Xu, Weiyao Wang, ..., Kevin J. Liang. Multi-SpatialMLLM: Multi-Frame Spatial Understanding with Multi-Modal Large Language Models. \
> [6] Sihan Yang, Runsen Xu, ..., Jiangmiao Pang. MMSI-Bench: A Benchmark for Multi-Image Spatial Intelligence. \
> [7] Linjie Li, Mahtab Bigverdi, ..., Ranjay Krishna. Unfolding Spatial Cognition: Evaluating Multimodal Models on Visual Simulations.

---

> > ### Comment · Reviewer_wMQv · 2025-08-06
> > **rebuttal**
> >
> > I appreciate authors detailed rebuttal, additional experiments and positioning the work with respect to the other references.
> > I also gained better appreciation of the value of comparison with human attentional salience.  I still have some doubts
> > when it comes to comparison and positioning of the work with respect to traditional VQA style evaluation on MLMMs and understanding of the failure modes. In typical VQA evaluations can quantify various linguistic properties of the models (lack of grounding of nouns, attributes, referring expressions) in addition to detections only as pointed out in the response to wr1L. VQA style evaluations get to the core of language priors, linguistic ambiguities, grounding of nouns, attributes, referring expressions etc. and typically bring these insights to the methods how these models are trained, suggest fine-tunning etc.
> >
> > I do appreciate the proposed spatial representations and new evaluations metrics and support the hypothesis
> > of SGI grids as representations.  Despite additional evaluations  I still find the connection between MLMM's weak, better discussion of generalization can be included.

---

> > > ### Author Response · Authors · 2025-08-06
> > > **Complementary relationship between SIG and text in representing visual-spatial intelligence**
> > >
> > > Thank you for your thoughtful feedback and appreciation of our contents in rebuttal regarding to addtional experiments and positioning of our work. We would like to further clarify the complementary role of SIG and traditional VQA-style representations in the context of visual-spatial intelligence (VSI).
> > >
> > > Traditional VQA evaluations typically encode spatial relations implicitly through natural language, which may require long and potentially ambiguous sentences to express spatial relationship between multiple objects. For example, to describe a relatively simple scene involving three cars and two traffic signs, one might need a sequence of sentences like: "The red car is to the left of the black car, and in front of the blue car. The nearest sign is behind the blue car and to the right of the black car...". Such textual descriptions can be both verbose and imprecise, even exceeding the token limit of MLLMs, especially when scaling up to complex scenes with many entities. On the other hand, SIG provides a more compact, structured, and precise representation of spatial relationships between all objects in the scene, encoding both directional (e.g., front, front left) and proximal (e.g., close to, far away from) relations.
> > >
> > > We agree with you that SIG cannot replace text-representation. Actually, we regard SIG as a complementary element to text for the improvement of VSI. As we mentioned in our work, we provide the textual representation of SIG named Spatial Relational Paragraph (SRP) and one of the task in our proposed benchmark is Spatial Relational Paragraph Filling (SRPF, see Fig. 5(b) for details), which is actually testing the MLLMs' ability to fill in prepositions between object names in the sentence (text format). Based on our knowledge, we think one of the best representation of VSI for now would be the combination of SIG and text, in which SIG captures structured spatial priors that are difficult to express concisely in text, especially in dense, dynamic environments like autonomous driving and text provides the fine-grained linguistic description that SIG is limited.
> > >
> > > We will definitely include this discussion in the final version of this paper and highlight the complementary relationship between SIG and text. We appreciate your comments that really strengthen our paper and we would be glad to engage in further discussion and address any additional questions you may have.

---

> ### Author Response · Authors · 2025-08-03
> **Update the complete results of SIG-COCO and SIG-ARKitScenes and results of new data sample selection strategy for SIG-based ICL**
>
> **W-1&Q-2 This is likely applicable only for AD scenarios.** We update the complete results of zero-shot and 3-shot ICL on SIG-COCO and SIG-ARKitScenes for general VSI tasks using GPT-4o as follows:
>
> | Benchmark         | Type    | **MLSM** |    |     |       | **SRGS** |     | **SRD (Directional)** |      |      | **SRD (Proximal)** |      |      |
> |-------------------|---------|:--------:|:--:|:---:|:-----:|:--------:|:---:|:---------------------:|:----:|:----:|:------------------:|:----:|:----:|
> |                   |         | P↑       | R↑ | F1↑ | AssA↑ | S↑       | WS↑ | MAE↓                  | MSE↓ | Acc↑ | MAE↓               | MSE↓ | Acc↑ |
> | SIG-COCO          | Z-S     | 0.758    |0.826|0.771|0.652  |0.574     |0.607|1.303                  |3.403 |0.407 |0.893               |1.160 |0.230 |
> |                   | ICL-SIG | 0.860    |0.840|0.849|0.749  |0.711     |0.746|0.713                  |1.827 |0.640 |0.357               |0.360 |0.643 |
> | SIG-ARKitScenes   | Z-S     | 0.714    |0.756|0.727|0.581  |0.619     |0.597|1.267                  |3.467 |0.433 |0.900               |1.167 |0.233 |
> |                   | ICL-SIG | 0.809    |0.850|0.823|0.719  |0.729     |0.737|0.467                  |0.533 |0.567 |0.300               |0.500 |0.800 |
>
> Please refer to our new comment to reviewer **BBAE W-2** for the new results on SIG-based ICL with another data sample selection strategy, which highlights the potential of SIG as a new data schema for improving VSI of MLLMs.

---

> > ### Author Response · Authors · 2025-08-06
> > **Happy to hear from you and discuss**
> >
> > Dear Reviewer wMQv,
> >
> > We appreciate you bringing the referenced work to our attention and it is helpful! Your feedback has clearly strengthened our work!
> >
> > We have carefully reviewed the paper, as well as the works cited by this paper. We note that our related work part does miss of mentioning some elements of visual-spatial intelligence (VSI) in existing works such as map-based representation and we will include them in the final version. Compared to all existing works we have looked through, our work is the first one that propose SIG as a novel data representation, design SIG-based evaluation metrics and propose SIG-based ICL. We demonstrate the performance and potential of SIG-based ICL through extensive experiments (You can refer to our latest comment to reviewer **BBAE** for the comparison of multi-run error bar statistics between text(multiple-choice vqa)-based ICL and SIG-based ICL on both GPT-4o and Gemini-2.5-Pro).
> >
> > Also, to better support future works for other researchers, we have confirmed with U.S.DOT and decided to release the SIGBench and SIGBench-tiny dataset, including original driving scenes images, images annotated with bounding boxes of vehicles, human-annotated SIG, SRP template with answers and human attention for each image. We believe our benchmark will provide a new persecptive in assessing the VSI of MLLMs and the first benchmark to evaluate it including human-like attention.
> >
> > We look forward to hearing your feedback and, we will be more than happy learn from your comments and thoughts!
> >
> > Sincerely,
> > Authors

---

### Note · Authors · 2025-08-12

Dear AC(s) and NeurIPS Committee,

Thank you for your effort for handling this submission and we appreicate all insightful feedbacks from all reviewers. Below we summarize the strengths acknowledged by reviewers and their concerns resolved during rebuttal.

**Strength**
- Our proposed benchmark and evaluation metrics for visual-spatial intelligence (VSI) based on our proposed new representation spatial intelligence grid (SIG) are **novel** (wMQv, BBAE), **thoughtful and well-motivated** (wr1L), addressing a **high importance and outstanding issue** in MLLMs (zSnK).
- We conduct **extensive experiments** (zSnK, wr1L, BBAE) and provides empirical evidence that SIG is **more effective for improving VSI** than standard VQA through in-context learning (ICL) (wr1L, BBAE).
- The integration of human attention provides **valuable insights** (wr1L, BBAE).

**Addressed concerns**
- **Generalization of SIG**: We create SIG-COCO and SIG-ARKitScenes, based on subset from MS COCO and ARKitScenes. Their results demonstrates generalizability of SIG beyond autonomous driving (wMQv, wr1L).
- **Runtime complexity**: We provide time complexity analysis and empirical results of our evaluation metrics, which meets real-time demand (wr1L).
- **Failure modes**: We do quantitative analysis (performance v.s. object count) and qualitative analysis: small/remote objects and high-IoU  (wr1L).
- **Human baseline, grid size of SIG**: We clarify gird-size rationale and highlight human basline (zSnK).
- **Data sample selection in ICL**: We provide detailed experimental results of performance using samples with: 1) different number of objects; 2) same image with different prediction. We also provide error bar statistics of 5-run ICL comparison using Gemini-2.5-Pro and GPT-4o, further illustrates the robustness of SIG-based ICL compared to VQA-based ICL (BBAE).
- **ICL on open-source MLLMs.**: We conduct SIG-based ICL on Qwen-VL-2.5 7B and 32B, which highlights its generalizability improving VSI across different MLLMs (BBAE).
- **Human-like VSI evaluation metrics**: We provide more detailed insight for the relationship between human saliency prediction and human-like VSI and provide the results of attention-based MLSM (wMQv, BBAE).
- **Additional.** We will: 1) mention the complementary roles of VQA and SIG (wMQv); 2) integrate traffic lanes and illustrate limitations (zSnK); 3) provide better analysis of human-like VSI (BBAE) in final version.

Sincerely,

The Authors of 2806

---

### Decision · Program_Chairs · 2025-09-17

**Decision:**

Accept (spotlight)

**Comment:**

This paper proposes the Spatial Intelligence Grid (SIG) and SIGBench as a new framework for evaluating and improving visual-spatial intelligence in multimodal LLMs. The representation is well-motivated, the metrics are rigorous, and the benchmark is thoughtfully built with human gaze data.

Experiments on both proprietary and open-source models show consistent gains over VQA-style baselines, and the rebuttal addressed concerns on generalization and efficiency. AC thinks the work is novel, timely, and has a clear impact for safety-critical domains like autonomous driving. AC recommend acceptance.